# Seasonal and ontogenetic variation of whiting diet in the Eastern English Channel and the Southern North Sea

Charles-André Timmerman[ID]◎*, Paul Marchal◎, Margaux Denamiel‡, Clémence Couvreur‡, Pierre Cresson◎

HMMN, Centre Manche—Mer du Nord, Ifremer, Boulogne-sur-Mer, France

◎ These authors contributed equally to this work.
‡ These authors also contributed equally to this work.
* charles.andre.timmerman@ifremer.fr

## Abstract

An accurate description of trophic interactions is crucial to understand ecosystem functioning and sustainably manage marine ecosystems exploitation. Carbon and nitrogen stable isotopes were coupled with stomach content analyses to investigate whiting (*Merlangius merlangus*, Linnaeus, 1758) feeding behavior in the Eastern English Channel and Southern North Sea. Whiting juveniles and adults were sampled in autumn and winter to investigate both ontogenetic and seasonal changes. In addition, queen scallops (*Aequipecten opercularis*) samples were collected along with fish to be used as isotopic benthic baseline. Results indicated an ontogenetic diet change from crustaceans to fish and cephalopods. In autumn, $\delta^{15}N$ values generally increased with fish size while in winter, a decrease of $\delta^{15}N$ values with fish size was observed, as a potential result of spatial variation in baseline $\delta^{15}N$ values. In winter, a nutrient-poor period, an increase in feeding intensity was observed, especially on the copepod *Temora longicornis*. This study provides further insights into whiting trophic ecology in relation to ontogenetic and seasonal variations, and it confirms the importance of combining several trophic analysis methods to understand ecosystem functioning.

## Introduction

Since trophic interactions shape communities' structure, determining fish diet is a key prerequisite for a better understanding of ecosystem functioning [1, 2] and a sustainable management of marine ecosystems [3, 4]. Ecosystem functioning refers to biotic and abiotic processes that occur within an ecosystem and determine its structure and stability [5, 6]. Trophic interactions play a central role on ecosystem functioning, implying transfer of energy and nutrients between species [7] but also trophic cascades [1, 8], the understanding of which is necessary to gain knowledge on food webs structure and eventually inform ecosystem-based management [3, 4, 9].

Organism's energetic requirements differ according to their size. Ontogenetic dietary shifts, the changes in resource use over the lifespan of a consumer, are widespread for many taxa [10,

**Data Availability Statement:** All raw data, fish biometry, stable isotope ratios and stomach contents are freely available online in the Seanoe repository at https://doi.org/10.17882/72233

**Funding:** CAT received a PhD grant through a project funded by European Marine Fisheries Fund and France Filière Pêche (2017-2020 VARITROPH project; PI:PC). All coauthors are supported by grant from the French government, the Region Hauts-de-France and Ifremer under the framework of the project CPER MARCO 2015-2020 The funders played no role in the study design, data collection and analysis, decision to publish, or preparation of the manuscript.

**Competing interests:** The authors have declared that no competing interests exist.

11]. Consumers make trade-offs between benefits (*e.g.*, foraging) relative to associated costs including mortality [10]. These trade-off changes according to ontogeny depending on the balance between the risk of mortality due to predation and the benefits received from the resource. Several biotic (predation risk, competition, prey availability) or abiotic (habitat use) factors may be responsible of these variations [12]. In addition, species intrinsic factors, such as the gape size or swimming abilities promote these ontogenetic changes. A positive relationship between size descriptors (*e.g.* body or gape size) and trophic level is thus classically observed for fish species in marine environments [13, 14], even if unexpected results recently called for a better examination of the pattern underlying this relationship [15]. Similarly, seasonal variation of environmental (temperature, nutrients) and biotic factors (metabolism, feeding intensity, prey availability) can alter feeding patterns [16]. Both ontogenetic and seasonal changes in fish foraging patterns have important consequences for ecosystem structure, function and stability, calling for a better consideration of their magnitude [17].

The Eastern English Channel and Southern North Sea (EEC-SNS) is home to a rich and intensively used ecosystem, which has long supported a wide range of human activities, in a climate change context. Local impact of fisheries is particularly important [18]. The present study focuses on whiting (*Merlangius merlangus* (Linnaeus, 1758)), an important demersal species in the EEC-SNS ecosystem, both in ecological and economic terms [15, 19]. Whiting is thus known to have one of the highest trophic level within the EEC-SNS fish community [20]. Due to its high trophic position, whiting biomass production is fueled by both pelagic and benthic pathways [21], as demonstrated by stable isotopes analyses [22, 23], leading to important biomass values for this species. The diet of this high trophic level species is well investigated in many ecosystems because it predates on commercially exploited fish and invertebrates, and also since it can compete with other economically valuable gadoid species [24]. Most studies highlight the main consumption of fish and invertebrates (*e.g.* crustaceans, mollusks or annelids). Some of them report an ontogenetic change in whiting diet, with small individuals mostly consuming crustaceans, and larger whiting mostly consuming fish. This feeding pattern has been previously observed in other ecosystems, *i.e.*, the North Sea [24–26], Baltic Sea [24], or Celtic Sea [27, 28]. However, ecosystems are subject to different environmental and ecological conditions. All differences in prey availability, predator abundance, fishing pressure and abiotic features make differences between the ecological role and diet of *M. merlangus* in an ecosystem and its conspecific in other ecosystems. To our knowledge, studies that have investigated EEC whiting diets have not focused on ontogenetic and seasonal diet changes [29, 30]. The increased trophic importance of fish with ontogeny is consistent with outputs of an independent modeling study based on an EEC application of the OSMOSE ecosystemic model [15]. Except this modeling study, the trophic ecology of whiting in EEC-SNS was mainly studied by stomach content analyses, providing a short-term view of its diet [24, 25, 30]. The use of stable isotope analyses in addition to analyses of stomach contents helps to resolve some common biases related to the analysis of stomach contents.

When consuming a prey, a predator integrates the carbon and nitrogen isotopic ratios of its prey into its own tissues, with a difference called isotopic discrimination [31]. The muscle nitrogen isotopic ratio ($\delta^{15}$N), empirically enriched by ~3.4 ‰ per trophic level, is classically used as an estimator of the trophic position [32]. Muscle carbon isotopic ratio ($\delta^{13}$C) is less enriched (~1 ‰ per trophic level), allowing its use as tracer of the origin of food sources [33]. In the marine environment, $\delta^{13}$C is used to distinguish between benthic and pelagic sources because benthic producers are typically $^{13}$C-enriched relative to pelagic producers [34]. Unlike stomach content analyses, stable isotope ratios of an organism's tissues provide information on the time-integrated assimilated diet. Combining stable isotopes and stomach content

analyzes is thus recognized as a powerful approach to gain accurate and complementary vision of fish trophic patterns at different time scales [35–37].

The aims of this study are (1) to investigate whiting trophic ecology in the EEC-SNS and, (2) to study ontogenetic and seasonal feeding changes at two seasons, autumn 2017 and winter 2018, using stomach content and stable isotopes analyzes. We will in particular test the hypothesis that whiting trophic ecology changes according to both ontogeny, due to gape size and energy requirements increasing, and seasonal variations in environmental and biotic factors.

## Materials and methods

### Studied area and sampling

Whiting individuals were collected during two bottom trawl surveys conducted in the EEC-SNS. Sampling locations are presented in the map (Fig 1), and actual coordinates are presented in the supporting materials (S1 Fig and S1 Table). These surveys investigate most components of the ecosystem, from the abiotic environment to top-predators. The CGFS (Channel Ground Fish Survey) occurred in autumn 2017 and the IBTS (International Bottom Trawl Survey) occurred in winter 2018, both on board of the R/V Thalassa [38, 39]. Protocols of all surveys are currently being evaluated by the French research institute for exploitation of the sea (Ifremer) and are validated by the ICES IBTS International Group [40]. In addition, survey's PIs received training about animal well-being and ethics.

Following the standard IBTS protocol [40], whiting and queen scallops (*Aequipecten opercularis*) individuals were collected using a 36/47 Grande Ouverture Verticale (GOV) bottom trawl towed for 30 minutes during daylight at a constant speed of 4 knots. Immediately after trawling, all fish were sorted, identified, weighed, and measured. Whiting individuals covering all size distribution in the trawl were kept for further analyzes. Fish were immediately frozen

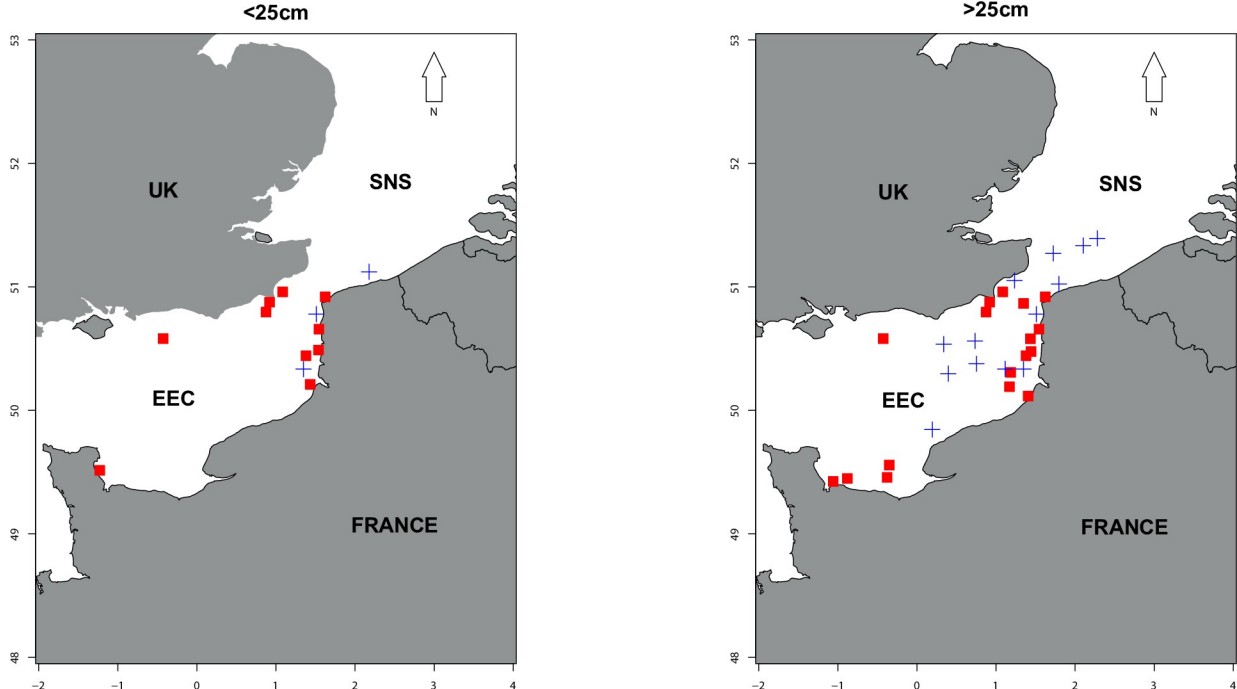

**Fig 1. Localization of stations sampled during the two surveys: CGFS 2017 (red squares, autumn 2017) and IBTS 2018 (blue cross, winter 2018).** Left: sampling locations of fish smaller than 25cm. Right: sampling locations of fish larger than 25cm.

(-50˚C) to stop digestion process, and then stored frozen onboard (-20˚C). Queen scallops (*A. opercularis*) samples were collected and stored frozen to be used as isotopic benthic baseline in the calculation of trophic level, following most classical approaches [20, 41, 42]. Organisms were sacrificed following the standard international IBTS protocol [40].

In the laboratory, fish were thawed and accurately measured (total length, to the nearest mm) before being dissected out, to collect muscle sample for isotopic analysis and stomach content. Gape size is an important parameter in the trophic relationships determinism and it was measured with truncated cones of different diameters. In order to cover the whole size range and to have a sufficient number of samples for numerical analyses, a minimum of 10 individuals per 5cm size classes were collected. A sample of white dorsal muscle (~ 2g wet mass) without skin (fish) or adductor muscle (bivalves) was dissected, stored frozen and then freeze-dried for at least 24h. White dorsal muscle is the most classical tissue used for stable isotopes analysis in fish because of its low turnover rate and low lipid content [43], notably for whiting, one of the species with the lowest C:N ratio and consequently lipid content in the area [44]. As C:N ratios measured in all samples were low (average value: 3.19 ± 0.08, min: 3.08, max: 3.43), raw isotopic ratios were used.

All samples (fish and *A. opercularis'* muscles) were ground into a homogeneous powder (~ 1g dry wet) with a mixer mill for isotopic analysis. A total of 248 individuals were sampled for stomach content analysis (124 in autumn and 124 in winter) and 212 were subsampled for stable isotopes (Table 1).

## Stomach content analysis

Prey items in stomach content were sorted under a binocular microscope and categorized into their lowest possible taxonomic group. The vacuity rate (%V) was used as an estimator of feeding intensity and it was calculated as the number of empty stomachs divided by the total number of stomachs. For each size class, dietary index was calculated based on Hyslop [45]. Frequency of occurrence ($O_i$) was first considered and calculated as the number of stomachs containing a prey type divided by the number of non-empty stomachs.

$$O_i = \frac{J_i}{P}$$

Where, $J_i$ is the number of fish containing prey $i$ and $P$ is the number of non-empty stomachs. This estimator is based on presence/absence, and it does not take into account the quantity of

**Table 1. Number of stomach content (n SCA) and stable isotopes (n SIA) analyzed by size classes, mean and range in stable isotopes ratios (mean δ¹⁵N ± SD, mean δ¹³C ± SD) observed and gape size for both seasons (autumn and winter).**

|        |          | n SCA | n SIA | δ¹³C (‰) mean ± SD | δ¹⁵N (‰) mean ± SD | C:N ratio mean ± SD | Trophic level mean ± SD | Gape size (mm) mean ± SD |
|--------|----------|-------|-------|---------------------|---------------------|----------------------|--------------------------|---------------------------|
| Autumn | 10-15cm  | 20    | 16    | -17.16 ± 0.93       | 14.42 ± 0.87        | 3.19 ± 0.08          | 3.86 ± 0.25              | 13 ± 2                    |
|        | 15-20cm  | 19    | 18    | -16.99 ± 0.41       | 14.97 ± 0.35        | 3.20 ± 0.06          | 4.06 ± 0.13              | 17 ± 2                    |
|        | 20-25cm  | 22    | 18    | -16.90 ± 0.47       | 16.02 ± 0.51        | 3.17 ± 0.08          | 4.40 ± 0.17              | 22 ± 1                    |
|        | 25-30cm  | 20    | 18    | -16.73 ± 0.30       | 16.33 ± 0.43        | 3.19 ± 0.09          | 4.40 ± 0.15              | 26 ± 3                    |
|        | 30-35cm  | 24    | 22    | -16.60 ± 0.34       | 15.95 ± 0.76        | 3.18 ± 0.06          | 4.33 ± 0.27              | 33 ± 2                    |
|        | 35+      | 19    | 15    | -16.65 ± 0.52       | 15.42 ± 0.73        | 3.18 ± 0.06          | 4.10 ± 0.30              | 41 ± 3                    |
| Winter | 15-20cm  | 33    | 28    | -17.31 ± 0.56       | 17.59 ± 0.89        | 3.19 ± 0.04          | 4.74 ± 0.26              | 19 ± 3                    |
|        | 20-25cm  | 23    | 21    | -16.77 ± 0.68       | 16.66 ± 0.12        | 3.20 ± 0.08          | 4.47 ± 0.25              | 22 ± 2                    |
|        | 25-30cm  | 23    | 21    | -16.77 ± 0.56       | 16.16 ± 0.98        | 3.23 ± 0.10          | 4.38 ± 0.28              | 28 ± 2                    |
|        | 30-35cm  | 31    | 21    | -17.20 ± 0.60       | 15.71 ± 0.85        | 3.22 ± 0.11          | 4.34 ± 0.23              | 33 ± 3                    |
|        | 35 +     | 14    | 14    | -17.34 ± 0.68       | 15.84 ± 1.12        | 3.20 ± 0.09          | 4.34 ± 0.27              | 39 ± 3                    |

prey items found in each stomach. For graphical representations, frequencies of occurrence were scaled to 100% and expressed as percentage.

$$\%O = \frac{O_i}{\sum O_i}$$

Percentage of abundance (%N) represented the number of prey items ($N_i$) relative to the total number of food items.

$$\%N = \frac{N_i}{\sum N_i}$$

Abundance and occurrence-based indices were used as they are considered less biased than indices calculated from prey biomass measurement [12, 45]. Assessing the occurrence or abundance of preys is a testimony of what was actually consumed, notably as prey abundance is based on a parsimonious method that limits overestimation of the number of degraded items, by estimating the most plausible number of prey individual that have originated the number of remains: paired items (*e.g.* otoliths, eyes, claws) were counted and divided by two. Other uncountable remains (*e.g.* muscles, gills, annelids bristles) were not counted and considered as remains of a single individual. On the contrary, biomass based indices are questioned, as they are more affect by well-known biases of stomach content. Differential digestibility of the preys leads to an overestimation of the mass of hard parts, and as biomass of parts found of the stomachs is the biomass of the undigested part of the diet, while stable isotopes reflect the part of the diet integrated in the tissues. Since stomach content analysis only provides a snapshot of fish diet, this approach was combined with stable isotopes analysis of fish tissue, which provides information on the time-integrated assimilated diet [35–37].

## Stable isotopes analyses

Isotopic analyzes were carried out with a Thermo Fisher Scientific Delta V Plus mass spectrometer (Bremen, Germany), coupled to a Flash 2000 elemental analyzer with a Conflow IV interface (Thermo Scientific, Milan, Italy). Ratios were expressed with the classical δ notation,

$$\delta X = \left( \frac{R_{sample}}{R_{standard}} - 1 \right) \times 10^3 \; (in \; ‰),$$

where X is $^{13}$C or $^{15}$N respectively, and R the ratio $^{13}$C/$^{12}$C or $^{15}$N/$^{14}$N respectively. Standards are Vienna Pee Dee Belemnite for $\delta^{13}$C and atmospheric nitrogen for $\delta^{15}$N. Accuracy of the analyses was checked by repeated measurements of internal acetanilide standards. The measurement error is less than 0.2‰ for both elements.

Since $\delta^{15}$N values result from the trophic position of the consumer, and from the isotopic ratio of the baseline, it is a relative proxy of the consumers' trophic position [32].The trophic level of whiting (TL$_i$) was calculated following Post [32] equation:

$$TL_i = TL_B + \frac{\delta^{15} N_i - \delta^{15} N_B}{TDF},$$

where TL$_B$ represents the trophic position of trophic baseline, $\delta^{15}$N$_i$ and $\delta^{15}$N$_B$ are the nitrogen isotopic ratios of fish and trophic baseline respectively, and TDF, the Trophic Discrimination Factor, represents the isotopic difference between a consumer and its diet. TL$_B$ was set to a value of 2, the theoretical trophic level of suspension-feeders, while TDF value was set to 3.4, following most classical approach [46, 47]. It is well known that isotopic ratios of the baseline vary spatially [41, 48, 49]. A station-specific baseline value was used to take into account the spatial

variability of isotopic ratios (S1 Fig and S1 Table). As *A. opercularis* samples could not always be found in the stations where fish were collected, missing $\delta^{15}N$ values were calculated with a geostatistical interpolation of baseline $\delta^{15}N$ values developed by Kopp et al. [29], and using *A. opercularis* isotopic ratios obtained in the present and previous published [41] and unpublished studies (P. Cresson unpubl. data). A total of 76 isotopic measurements were used as a baseline for geostatistical interpolation (mean $\delta^{15}N$: 7.69 ± 0.90 ‰). The dependency of covariance on distance between two sampling sites was modelled using a non-linear regression based on weighted least squares. The weight is the number of pairs of points per distance intervals. Spatial covariance in $\delta^{15}N$ was best described by a Gaussian model of distance with adjR2 = 0.99.

## Numerical analysis

The seasonal and ontogenetic variations of stable isotope values ($\delta^{15}N$, $\delta^{13}C$), abundances and occurrences of prey found in stomach contents, were investigated using Generalized Linear Models (GLM). Two discrete explanatory variables were thus considered: fish size (5cm size classes) and season (autumn and winter). Individual effects of these two explanatory variables, as well as their interactions were examined. Using discrete size classes in isotopic analyses GLMs was justified a posteriori by the non-monotonic character of the isotopic ratios versus size relationships. For stomach contents analyses, size-class analyses allow to account for the highly non-linear nature of dietary mechanisms. Each individual within a size class is considered as a replicate of the class. Having the diet of several individuals within a size-class is required to capture the diet of a class, and to limit the importance of extreme feeding mechanisms. Individual effects of these two explanatory variables, as well as their interactions were examined. The explained variables are structurally different, some are continuous (stable isotope values), others are counts (abundances) and eventually prey occurrences are binary variables, so different probability distributions were assumed to underlie their variations. For stable isotopes values, a Gaussian type was assumed, and the normality of residuals was tested by examining the characteristic Quantile-Quantile (QQ) plot [50]. For prey abundances and occurrences, GLMs were performed separately on the main categories of preys (*i.e.* fish, mollusks, benthic crustaceans, pelagic crustaceans and unidentified crustaceans). A binomial distribution with a logit link function was applied to model the variability of binary prey occurrence (presence = 1, absence = 0). Finally, a GLM building on a Poisson distribution and a log link function was applied to model the variability of prey abundance counts.

All analyses and figures were performed with R version 3.6 [51]. Maps were produced using akima [52], GISTools [53] and mapdata [54] packages. Statistical analyses were carried out using the car [55] package, using the glm function. Most figures were realized using ggplot2 [56].

# Results

Fish size ranged between 106 and 528 mm. Average size was similar at both seasons for both the whole dataset (246 ± 89 mm in autumn, 257 ± 76 mm in winter) and the subsample considered for stable isotopes (252 ± 83 mm in autumn, 262 ± 76 mm in winter). Subsample used for stable isotopes analyses can thus be considered representative of the larger sample used for stomach content analyses. Although the average size was comparable between the two seasons, no whiting smaller than 15 cm were collected in winter. Gape size increased with fish size and ranged from 13 ± 2 mm for the smallest size class to 41 ± 3 mm for the largest in autumn (Table 1).

## Stomach content analyzes

Vacuity rates were low for both seasons (16% in autumn and 7% in winter). Whiting diet was mainly composed of fish ($O_i$ = 0.57, %O = 25.5%), crustaceans, with a similar importance of

benthic ($O_i$ = 0.43, %O = 19%) and pelagic ($O_i$ = 0.38, %O = 17%) species, and mollusks ($O_i$ = 0.17, %O = 8%) (Fig 2).

Despite the degradation of fish prey, the main groups observed in stomach contents were perciforms in autumn (*e.g.* Callionymidae, Gobiidae, Carangidae) and clupeiforms (*e.g. Clupea harengus, Sprattus sprattus*) in winter. Similar benthic crustaceans were consumed in both seasons: peracarids (*e.g.* amphipods, cumaceous) for the smallest size classes and decapods for the

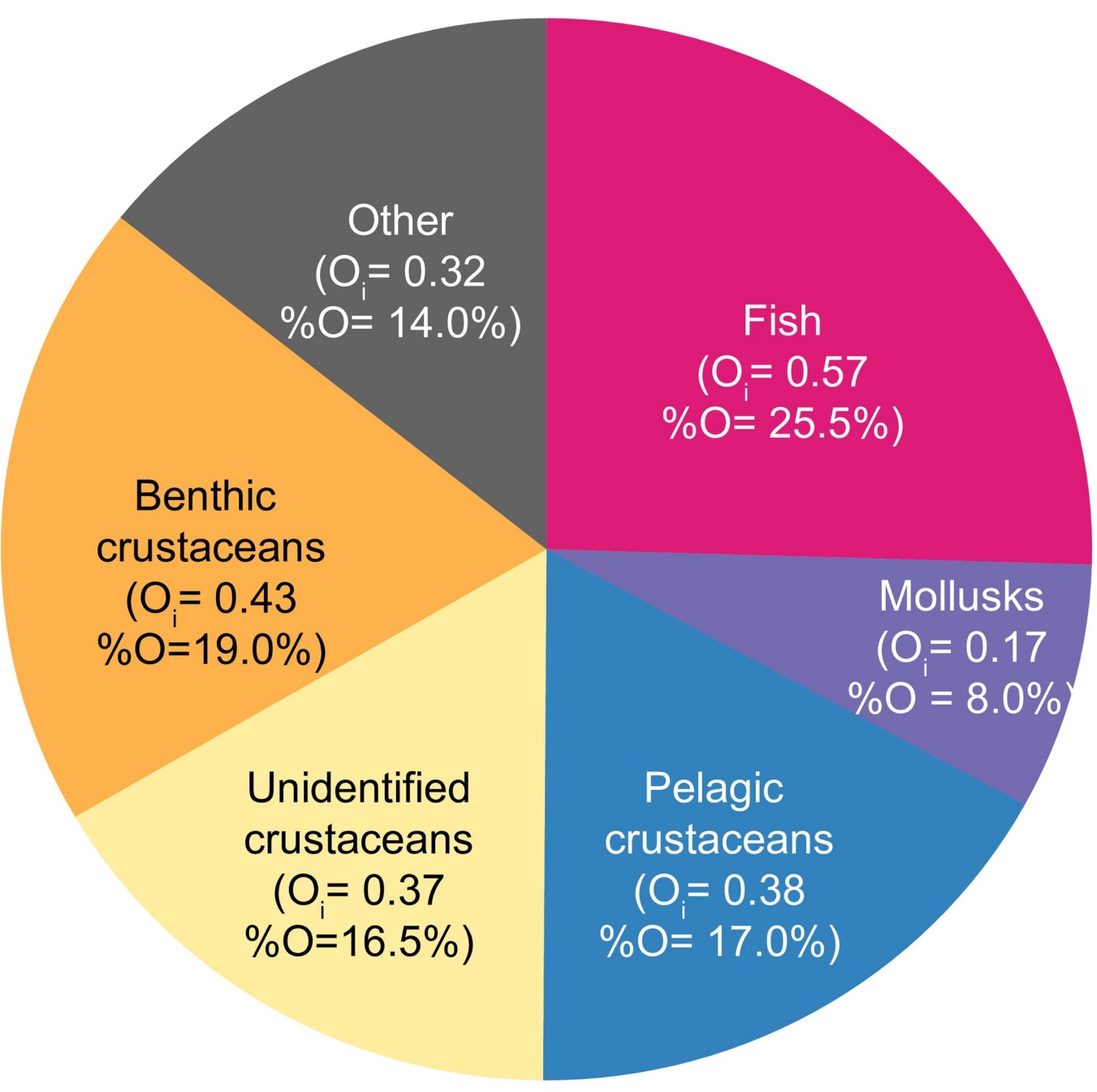

**Fig 2. Whiting diet averaged over seasons and size classes, expressed as frequency of occurrence ($O_i$).** Since a stomach may contain several preys, the sum of the frequencies of occurrence represents more than 1. For each prey group, frequencies of occurrence were scaled to 100% for the graphical representation (%O).

largest. Pelagic crustaceans were mainly copepods, with *Temora longicornis* being very abundant in winter (%N = 63%), and mysids. The taxon-specific results can be found in the dataset published at the following address: https://doi.org/10.17882/72233. For mollusks, bivalves and gastropods dominated diet of smallest individuals while the largest whiting mainly fed on cephalopods at both seasons. Among other prey categories, annelids, cnidarians, echinoderms or plant debris were also found in stomach contents at both seasons but in smaller quantities. They were then grouped into one group referred to as "other" hereafter.

The same categories of prey were found between autumn and winter but their relative importance differed according to seasons and size classes (Fig 3), whether these factors are considered individually or in interaction (Table 2). In winter, as whiting grows, it consumes more fish, both in occurrence and abundance, with the largest size classes (above 30cm) feeding more selectively on clupeids (Table 2 and Fig 3). A similar ontogenetic increase appears in the consumption of mollusks, both in abundance and occurrence, with little seasonal difference (Fig 3 and Table 2).

In contrast, the abundance and occurrence of crustaceans decreased according to ontogeny but patterns differed according to the nature of crustaceans (benthic, pelagic and unidentified) and seasons. For benthic crustaceans, the decrease in occurrence during ontogeny was mainly pronounced in winter (Fig 3 and Table 2). The decrease in abundance was observed for both seasons (p-value<0.001), but the highest prey abundance was observed for the 20-25cm size class.

The ontogenetic decrease in occurrence of pelagic crustaceans was similar in autumn and winter (Fig 3 and Table 2). Abundance decreased for both seasons according to size classes (Fig 3 and Table 2) but more pelagic crustaceans were consumed in winter. The high consumption of pelagic

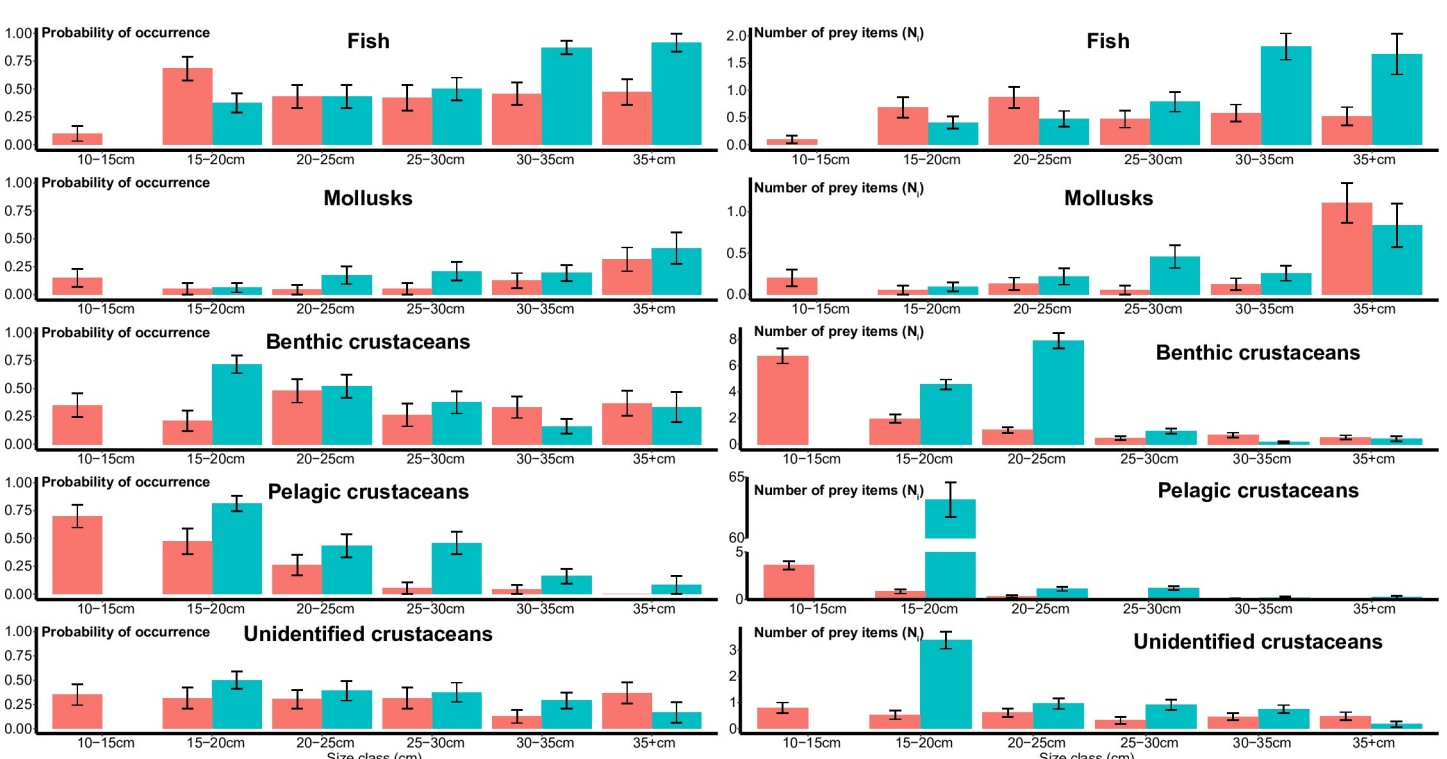

**Fig 3. Predicted values from GLM for average probability of prey occurrences (left) and number of prey items ($N_i$) in an individual stomach (right) and for both seasons (autumn in red and winter in blue).** Errors bars correspond to standard errors. For graphical purposes, the vertical axis of the plot presenting abundance of pelagic crustaceans was cut between 5 and 60cm.

Table 2. Results of generalized linear model (type 3).

| Response variable | df | Residual deviance | Explanatory variables | df | Deviance | Residual deviance | df | p-value (Chi-square) |
|---|---|---|---|---|---|---|---|---|
| **Occurrence–fish** | 245 | 291.28 | **Size class** | 5 | **26.55** | **314.42** | **240** | **<0.001** |
| | | | Season | 1 | 2.77 | 311.65 | 239 | 0.10 |
| | | | **Size class × season** | 4 | **20.37** | **291.28** | **235** | **<0.001** |
| **Occurrence - Benthic crustaceans** | 245 | 299.05 | **Size class** | 5 | **13.23** | **314.94** | **240** | **0.02** |
| | | | Season | 1 | 2.17 | 312.76 | 239 | 0.14 |
| | | | **Size class × season** | 4 | **13.72** | **299.05** | **235** | **0.009** |
| **Occurrence - Pelagic crustaceans** | 245 | 223.03 | **Size class** | 5 | **70.88** | **244.99** | **240** | **<0.001** |
| | | | **Season** | 1 | **18.48** | **226.50** | **239** | **<0.001** |
| | | | Size class × season | 4 | 3.74 | 223.03 | 235 | 0.48 |
| **Occurrence - Unidentified crustaceans** | 245 | 299.73 | Size class | 5 | 6.01 | 305.75 | 240 | 0.31 |
| | | | Season | 1 | 1.79 | 303.96 | 239 | 0.18 |
| | | | Size class × season | 4 | 4.24 | 299.73 | 235 | 0.38 |
| **Occurrence - Mollusks** | 245 | 190.13 | **Size class** | 5 | **12.86** | **195.46** | **240** | **0.02** |
| | | | Season | 1 | 3.80 | 191.66 | 239 | 0.05 |
| | | | Size class × season | 4 | 1.52 | 190.13 | 0.82 | 0.82 |
| **Abundance - fish** | 245 | 264.21 | **Size class** | 5 | **41.62** | **297.25** | **240** | **<0.001** |
| | | | **Season** | 1 | **9.49** | **287.76** | **239** | **0.002** |
| | | | **Size class × season** | 4 | **23.55** | **264.21** | **235** | **<0.001** |
| **Abundance - Benthic crustaceans** | 245 | 1888.00 | **Size class** | 5 | **475.28** | **2061.80** | **240** | **<0.001** |
| | | | **Season** | 1 | **110.95** | **1950.90** | **239** | **<0.001** |
| | | | **Size class × season** | 4 | **62.94** | **1888.00** | **235** | **<0.001** |
| **Abundance - Pelagic crustaceans** | 245 | 3871.60 | **Size class** | 5 | **5574.00** | **5644.40** | **240** | **<0.001** |
| | | | **Season** | 1 | **1735.00** | **3909.30** | **239** | **<0.001** |
| | | | **Size class × season** | 4 | **37.80** | **3871.60** | **235** | **<0.001** |
| **Abundance - Unidentified crustaceans** | 245 | 588.34 | **Size class** | 5 | **99.62** | **652.42** | **240** | **<0.001** |
| | | | **Season** | 1 | **41.84** | **610.58** | **239** | **<0.001** |
| | | | **Size class × season** | 4 | **22.24** | **588.34** | **235** | **<0.001** |
| **Abundance - Mollusks** | 245 | 235.12 | **Size class** | 5 | **48.69** | **245.29** | **240** | **<0.001** |
| | | | Season | 1 | 1.96 | 243.33 | 239 | 0.16 |
| | | | Size class × season | 4 | 8.21 | 235.12 | 235 | 0.08 |
| $\delta^{13}C$ | 211 | 64.26 | **Size class** | 5 | **5.64** | **72.76** | **206** | **0.003** |
| | | | **Season** | 1 | **4.04** | **68.72** | **205** | **<0.001** |
| | | | **Size class × season** | 4 | **4.46** | **64.26** | **201** | **<0.001** |
| $\delta^{15}N$ | 211 | 126.84 | **Size class** | 5 | **67.12** | **208.22** | **206** | **<0.001** |
| | | | **Season** | 1 | **23.58** | **184.64** | **205** | **<0.001** |
| | | | **Size class × season** | 4 | **57.80** | **126.84** | **201** | **<0.001** |
| **Trophic level** | 211 | 11.50 | **Size class** | 5 | **5.36** | **206** | **206** | **<0.001** |
| | | | **Season** | 1 | **1.94** | **205** | **205** | **<0.001** |
| | | | **Size class × season** | 4 | **3.60** | **201** | **201** | **<0.001** |

df: Degrees of Freedom. Significant effects (at p-value<0.05) of the tested explanatory variables are highlighted in bold characters.

crustaceans in winter (primarily the copepod *T. longicornis*) mainly concerned the 15–20 size class. Pelagic crustaceans were 21 times more abundant in stomachs in winter than in summer.

Finally, occurrence of unidentified crustaceans did not vary according to size class nor season, while abundance decreased with ontogeny, especially in winter (p-value<0.001, Table 2) similarly to the two other crustacean groups.

### Stable isotopes

$\delta^{15}N$ values varied within a 2‰ range for both seasons, from 14.42 ± 0.87‰ (10-15cm) to 16.33 ± 0.43‰ (25-30cm) in autumn, and from 15.71 ± 0.85‰ (30-35cm) to 17.58 ± 0.78‰ (15-20cm) in winter (Fig 4 and Table 2). Trophic levels ranged from 3.86 ± 0.25 (10-15cm) to 4.40 ± 0.15 (25-30cm) in autumn, and from 4.34 ± 0.27 for largest individuals (>35cm) to 4.74 ± 0.26 for smallest individuals (15-20cm).

However, slight differences according to seasons and size class were observed (Fig 4 and Table 2). Size effect on both trophic level and $\delta^{15}N$ values differed between seasons for smaller whiting (<25cm), with an ontogenetic increase in autumn. In winter, the unexpectedly high value measured for the smallest individuals led to a general decreasing trend. Size effects were season-invariant for larger whiting (>25cm), with a similar slight ontogenetic decrease in both autumn and winter. $\delta^{13}C$ values displayed approximatively a 0.5‰ range for both seasons and varied significantly (p<0.001, Table 2) from -17.16 ± 0.93‰ (10-15cm) to -16.60 ± 0.34‰ (30-35cm) in autumn and from -17.34 ± 0.68‰ (35+) to -16.77 for both 20–25 (-16.77 ± 0.68‰) and 25–30 (-16.77 ± 0.65‰) size classes in winter. In autumn, $\delta^{13}C$ increased with fish size, while in winter it increased up to 20cm and decreased after 30cm (Fig 4).

## Discussion

### General ontogenetic patterns in whiting diet

*M. merlangus* is considered as a piscivore species, notably for larger individuals [24, 26, 57]. Consistently with previous studies [24, 26, 27], in the EEC-SNS, piscivory mainly concerned largest size classes. The increased trophic importance of fish with ontogeny is also consistent

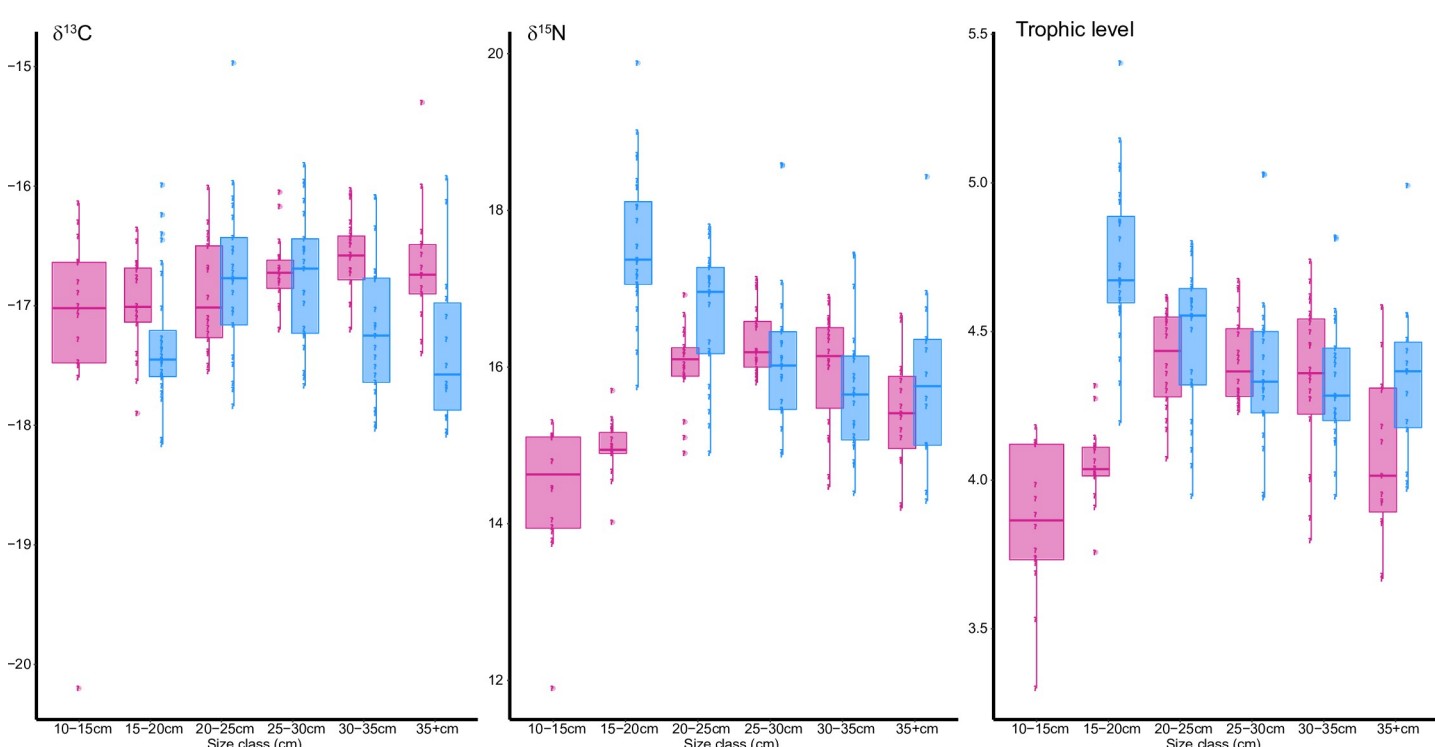

**Fig 4. Relationship between $\delta^{13}C$, $\delta^{15}N$ and trophic level according to size classes in autumn and winter respectively (autumn in red and winter in blue).** The edges of the boxes are the first and third quartiles, the horizontal lines are median values. Outliers are calculated using 1.5 times interquartile space (distance between the first and third quartile).

with outputs of an independent modeling study based on an EEC application of the OSMOSE ecosystemic model [15]. Interestingly, outputs from this model already predicted the predominance of fishes in whiting diet for individuals of 25cm or more. However, fish do not constitute the main food item for smallest size classes. Smallest size classes mainly consumed pelagic crustaceans. A similar pattern was previously observed for saithe (*Pollachius virens*), another gadoid species with a strong specialization of smaller individuals on euphausiids and copepods [58].

Most studies highlighted the consumption of crustaceans but did not distinguish between benthic or pelagic species [24, 26, 27]. However, this consideration is important for a better understanding of food web functioning and fluxes. Previous studies based on stable isotopes confirmed the important benthic-pelagic coupling in this area, but were not able to identify mechanisms at play, *i.e.*, if coupling results from the consumption of benthic invertebrates by pelagic fish or from the integration of benthic production in the pelagic pathway [22, 29]. The similar importance of both groups of crustaceans may testify that the two hypotheses cannot be ruled out.

## Seasonal ontogenetic patterns in whiting diet

**Autumn.** Results obtained from stable isotopes analyzes in autumn are similar with those obtained by Kopp et al [29] during the same period, in a different year. In the present study, results obtained in autumn from stomach content analyses and carbon isotopes values displayed consistent patterns. Since pelagic organisms are generally $\delta^{13}C$-depleted compared to benthic ones [29, 34] the increase of $\delta^{13}C$ values is thus consistent with a diet switch from pelagic invertebrates to benthic fish as whiting grows. The link between prey found in stomach contents and $\delta^{15}N$ values is of a more complex nature though. The increase of $\delta^{15}N$ values could reflect the ontogenetic diet shift, from crustaceans to higher trophic level cephalopods, as revealed by stomach contents analyzes, but only for whiting ranging between 15 and 30cm. This increase is an expected pattern and it was observed in many ecosystems and for many taxa [10, 59, 60]. The positive relationship between $\delta^{15}N$ values and body size is commonly observed for piscivores [13]. Larger fishes have a larger gape size, but also higher energetic demands. Consumption of larger, energy-richer and of higher trophic level prey becomes an obvious way to fulfill metabolic needs [59]. Ontogenetic diet shift is well-documented for whiting in other ecosystems, *i.e.*, in the North Sea [24, 26, 43], Celtic Sea [27, 28], or Baltic sea [24] and generally occurs between 15 and 30cm. Ontogenetic shift probably occurs due to increase of gape size and enhanced detection and capture abilities allowing larger individuals to maximize their energy input through the consumption of larger prey. Fish do have a higher energetic content than invertebrates [61]. Similarly, even if the energetic content per unit of body mass is ~10 times higher for bivalves than for cephalopods [62, 63], cephalopods are heavier than bivalves, and might also be more accessible than shell-sheltered organisms, despite their higher mobility. The consumption of a cephalopod can thus be more profitable in terms of energy input.

It is not entirely clear why $\delta^{15}N$ values decrease slightly with fish size for larger whiting (>30cm) but the decrease is low (~1‰), *i.e.*, of ~0.3 trophic level assuming a trophic discrimination factor of 3.4‰ [32] and may not track a change in diet. The study site is shallow and presumably has high $\delta^{15}N$ value, because denitrification in sediments increase the $\delta^{15}N$ value of pore water, and consequently of the organic matter at the base of food webs [64, 65]. Therefore, because whiting migrate only after the first year of life [66], smaller fish with low migration ability had higher $\delta^{15}N$ values. On the other hand, since isotopic ratios of the baseline vary spatially [41, 48], we can hypothesize that the decrease in $\delta^{15}N$ for larger individuals does not reflect a different diet, but an artifact related to a spatially different baseline. As mentioned by

Reddin et al. [49], from an isotopic point of view, baseline can sometimes be more variable than the diet. Here, the difference between the minimal and maximal $\delta^{15}N$ values kriged for queen scallops was of ~1.5‰ (S1 Table). However variations may be more important if considering neighboring areas, *i.e.* the Western English Channel or northern areas in the North Sea [41, 48]. Assuming high migration abilities for larger whiting [66], their isotopic ratios may reflect a diet in areas where baseline $\delta^{15}N$ is markedly lower. Investigating isotopic ratios of whiting over a large spatial continuum, with samples from the whole English Channel and the North Sea could thus be useful to better understand the influence of spatial migration on fish isotopic ratios.

**Winter.** At the beginning of the winter period, higher prey abundances in stomach contents and lower vacuity rates evidenced an increase of prey uptake and feeding intensity. This was particularly reflected by a high abundance of *T. longicornis* in 15-20cm whiting stomachs. The large consumption of pelagic crustaceans, especially copepods, was also reported in the North Sea [25, 67]. This increase may result from an opportunistic behavior to supply energetic demands during a nutrient-poor period, for a size class that is constrained in the range of prey it can access and feed on. In the EEC-SNS, winter nutrient concentration and temperatures were lower than in autumn [68–70]. All copepods do not have the same behavioral response to temperature changes. For instance, *T. longicornis* slows down its swimming speed and activity during colder conditions [71] while *Calanus finmarchicus* is not affected by changing temperature conditions [72]. Consequently, increasing the consumption of *T. longicornis* in winter would be favored, as catching this species would come at relatively low energy cost. Furthermore, *T. longicornis* has a relatively low energetic value per individual [73], which may explain why it was found in large quantities to fulfil whiting energetic needs. $\delta^{13}C$ values obtained in winter are consistent with a diet based on pelagic crustaceans for small size class, then switching to benthic crustaceans for intermediates and pelagic fish for largest individuals. Past [19] and present results confirm that Clupeidae are an abundant and accessible prey for large whiting in winter. During this period, Downs herring [74] and *S. sprattus* [75] gather in very large numbers in the EEC-SNS to mate and spawn. The large abundance of Clupeidae in large whiting stomachs reflects that the winter spawning migration of these species in the EEC represents an important trophic opportunity for whiting during a nutrient-poor period. Previous studies reported a particularly high energetic value for these species [63, 76]. Energy content analyzes performed on 78 species taken from the northeast Atlantic Ocean thus revealed that *C. harengus* was the prey with the highest energy density [63]. Nevertheless adult sprats (maturity between 7 and 14cm, [69]) are smaller than adult herring (maturity between 20 and 30cm, [69]), making them more accessible for large whiting despite their lower energy content.

However, the overall decrease of $\delta^{15}N$ and trophic level with fish size observed in winter is inconsistent with diet, highlighting the interest of coupling stomach content and stable isotopes analyses. This negative trend was observed for some benthic species, due to shifts in prey preferences, from carnivore benthic species to suspension-feeders prey [15, 29, 77]. However, the relative nature of stable isotopes values calls for a cautious interpretation of odd and unexpected patterns. This trend may be largely driven by unexpectedly high trophic levels (4.74 ± 0.26) calculated for small whiting individuals. Lower trophic levels (~2.6–3.1) were for example estimated by Jennings and van der Molen [20] for individuals of the same size. Thus, the capacity of $\delta^{15}N$ values to reflect trophic level could be blurred by inaccurate baseline or fractionation factors [37, 78]. Previous studies notably highlighted the importance of spatial factors, here the inshore-offshore gradient on the food web structure and consequently on species' isotopic ratios in the English Channel [20, 29]. Ontogenetic change in diet, as already discussed, and the predominance of pelagic crustaceans in the diet of smallest whiting suggest that small whiting do not feed on high trophic level prey in winter. Interestingly, small whiting

(<20cm) sampled in January 2017 (*i.e.* one year before present sampling) at coastal locations close to the stations where small whiting were collected in the present work exhibited similarly high $\delta^{15}$N values (P. Cresson, unpubl. results). Thus, several hypotheses can be proposed to explain these unexpectedly high $\delta^{15}$N values consistently found in small whiting.

One of the well-known limitations of stomach contents is the impossibility of identifying soft-bodied prey, such as fish eggs, due to their rapid digestion [45, 79]. Fish eggs may have high $\delta^{15}$N values, sometimes even higher than their mother [80]. The nitrogen pool present in eggs, and in particular in the yolk sac, is inherited from their parents [80]. A previous study measured high $\delta^{15}$N values (~10 to 17‰) for Downs herrings caught in their feeding grounds in the northern North Sea, *i.e.*, in a remote location where baseline $\delta^{15}$N value may be well higher than in the SNS [48]. We then first hypothesized that increased $\delta^{15}$N values in small whiting might result from the consumption of eggs laid by Downs herring females that fed in the northern feeding ground. Since fish eggs are lipid-rich [81], their consumption would be largely beneficial, notably as eggs are abundantly laid on the bottom by spawning herring females. This could represent a major though very limited in time bonanza, as already observed for other species and in other environments [81, 82]. Nevertheless, preliminary results obtained on three females and their eggs sampled in November 2019 in the EEC did not allow confirming this hypothesis, since isotopic ratios measured in herring muscle ($\delta^{15}$N = 11.67 ± 1.33‰) and eggs (11.44 ± 0.58‰) were not high enough to explain nitrogen enrichment in small whiting (P. Cresson, unpubl. data). Future studies based on a larger sample size, and including a simultaneous collection of herring adults, juveniles and eggs are nonetheless needed to ascertain the consumption of eggs by small whiting, and to exclude interannual variability in the isotopic ratios of herring in the EEC-SNS.

Another possible explanation of these high $\delta^{15}$N values in small whiting can be linked to local specificities. Small whiting were collected in winter at very coastal locations, close to the outfalls of rivers Somme (in the bay of Somme) and Aa, off the city of Gravelines (Fig 1). Catchments of these two rivers cross zones of intense agricultural activity, and pass along mid-populated cities [83]. Several papers already demonstrated that the integration of runoff from agricultural or urbanized areas leaf to increase nitrogen isotopic ratio in consumers [84]. Jennings and van der Molen [20] also estimated high $\delta^{15}$N values for coastal queen scallops, as a result of a positive relationship between estimated $\delta^{15}$N and minimum bottom salinities.

In addition, Gravelines hosts a fish farm (https://www.gloriamarisgroupe.com/aquanord/?lang=en), raising 1 800 tons per year of gilthead sea bream (*Sparus aurata*) and sea bass (*Dicentrarchus labrax*). Integration of fish farm effluents may have also increased the $\delta^{15}$N values of the surrounding food webs, notably as a result of the integration of fish feces [85]. Isotopic ratios of raised fish are higher than wild counterparts [86]. Here, the integration of feces derived from predatory fishes may locally increase $\delta^{15}$N baseline values, and explain higher $\delta^{15}$N values of small whiting observed in the present study. Unfortunately, no information on isotopic ratios of coastal queen scallops could be collected as the survey trawl could not be operated in shallow waters. Thus, as kriging is directly dependent upon input data, local influences on queen scallops isotopic ratios could not be efficiently captured, and may have induced an overestimation of small whiting $\delta^{15}$N values and trophic levels. Interestingly, estimation of trophic level was associated with highest uncertainties at coastal locations in the model of Jennings and van der Molen [20], as a result of the effect of salinity on baseline $\delta^{15}$N ratios. The high concentration of juveniles in coastal areas, and their limited movement abilities may explain why this influence of terrigenous or fish farm derived nitrogen would primarily affect the smallest size class.

Stomach content analysis provides short-term information about the integrated diet, from hours to days [45, 87]. Stable isotopes analysis provides information on assimilated diet at a longer time scale than stomach contents, but varies according to the lifecycle of species. A

meta-analysis carried out on several taxa revealed that, in general, turnover rate differs between juveniles and adults. Stable isotopes ratios change due to growth and metabolic replacement associated with a change in diet were observed between juveniles and adults [88]. These authors have developed an equation to estimate the turnover rate according to the body mass (proxy of the size) and to the type of species (endotherm vs. ectotherm) and tissue. Tissues characterized by a quick turnover provide the most recent dietary information. Using this equation, the turnover rate was estimate to ~45 days for juveniles (15g average mass) which is consistent with the turnover rate estimated for cod, another gadoid species [89]. We then admit that juveniles 'isotope ratios may reflect feeding during the collection season (autumn or winter). For adults (weighing 508g on average), the turnover rate was of about three months, consistently with the expected turnover of adult fish [87, 90, 91]. Measured isotopic ratios may partly result from dietary elements integrated in the tissues before the sampling season, and away from EEC, due to movement abilities of adult whiting [66]. However, isotopic integration of diet has to be viewed from a dynamic point of view. Addressing the seasonal (*i.e.* in autumn and winter) and spatial (*i.e.* over the potential area covered by whiting over a three months period) variability of prey isotopic ratios would also be needed. Finally, since the integration of isotopes in tissues is dynamic, additional studies are necessary to know precisely the rate at which isotopes are incorporated into whiting's muscle and the discrimination factor between tissues and diet [92]. It should also be noted that the method used to estimate turnover rates does not distinguish between isotopes ($\delta^{13}$C, $\delta^{15}$N). However, several studies have shown that for fish, the turnover rate of $\delta^{13}$C is faster than that of $\delta^{15}$N due to different catabolic processes operating on the various biochemical constituents of tissues (amino acids, proteins, lipids [93–95]). Consequently, both $\delta^{13}$C and $\delta^{15}$N for the same species inform about diet at different time scales, highlighting the importance of considering the turnover rate to know when that diet has been assimilated for a better understanding of species diet.

This study has led to a better understanding of whiting trophic ecology. Despite slight variations according to seasons, results obtained for the autumn period are consistent with previous results obtained from stable isotopes and stomach content analyzes in autumn in the same area but in different year [29, 30], highlighting trophic consistency through years for this species. Results obtained from both stable isotopes, and stomach content confirmed that the high trophic position of whiting allows this species to benefit from benthic and pelagic pathways in the EEC, explaining its high biomass [21]. The confirmation of the ontogenetic change of diet for whiting confirms the need for a better consideration of this aspect of the dietary changes in this species [17].

Several research perspectives emerge from this study. First, although our work focused on two seasons (autumn and winter), future studies could extend these analyzes to spring and summer. Higher temperature and productivity during these seasons may thus drive different ecosystem functioning patterns and trophic behaviors. Second, one could investigate the major effects of intraspecific diet variations on ecosystem functioning, which are commonly overlooked [96, 97]. In addition, this study considered 5cm-size class as homogeneous but discrepancies may also occur between individuals of the same class. Future studies could investigate these variations, and their potential effect on ecosystem functioning. The effects of ontogenetic variation could also be integrated in ecosystem models, such as those currently being implemented in the EEC, *e.g.*, Atlantis [19], or OSMOSE [15], thereby increasing their capacity to capture the complexity of marine ecosystems and to inform fisheries management [4].

## Supporting information

**S1 Fig. Localization of whiting (orange crosses) and *Aequipecten opercularis* (green crosses) sampling stations by this, but also previous studies (Jennings & Warr 2003, P.**

Cresson unpubl. data). An interpolation of *A. opercularis*' $\delta^{15}$N values was performed to obtain baseline isotope ratios at all stations where whiting were collected (see the method in the material and method section of the manuscript).
(DOCX)

**S2 Fig. Ontogenetic dietary changes for autumn and winter.** Figures A and B represented percentages of occurrence (%O) in autumn (A) and winter (B) respectively. Figures C and D represented percentages of abundance (%N) in autumn (C) and winter (D) respectively. The number of non-empty stomach content is expressed under each size class.
(DOCX)

**S1 Table. Geographic coordinates of the whiting sampling stations for both seasons (CGFS in autumn and IBTS in winter).** HaulNum corresponds to the number of the sampling station. $\delta^{15}$N baseline corresponds to interpolated $\delta^{15}$N isotopic ratios of *A. opercularis* at all stations where whiting were collected.
(DOCX)

## Acknowledgments

We are grateful to the technical and scientific crews of R/V "Thalassa" for their work during CGFS 2017 and IBTS 2018 surveys, to L. Millet who dissected fish and prepared samples for isotopic analyzes, to the team of the Plateforme Spectrométrie Océan who performed isotopic analyzes. We are also grateful to two anonymous reviewers for their insightful comments on a previous version of this manuscript.

## Author Contributions

**Conceptualization:** Charles-André Timmerman, Paul Marchal, Pierre Cresson.

**Data curation:** Charles-André Timmerman.

**Formal analysis:** Charles-André Timmerman, Paul Marchal, Pierre Cresson.

**Funding acquisition:** Pierre Cresson.

**Investigation:** Charles-André Timmerman, Margaux Denamiel, Clémence Couvreur.

**Methodology:** Charles-André Timmerman, Paul Marchal, Pierre Cresson.

**Project administration:** Paul Marchal.

**Resources:** Margaux Denamiel, Clémence Couvreur, Pierre Cresson.

**Supervision:** Paul Marchal, Pierre Cresson.

**Validation:** Charles-André Timmerman, Paul Marchal, Margaux Denamiel, Clémence Couvreur, Pierre Cresson.

**Writing – original draft:** Charles-André Timmerman, Paul Marchal, Pierre Cresson.

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
