## [Decision Letter · Decision Letter 0]

21 Apr 2020

PONE-D-20-06925

Seasonal and ontogenetic variation of whiting diet in the Eastern English Channel and the Southern North Sea

PLOS ONE

Dear Mr. Timmerman,

Thank you for submitting your manuscript to PLOS ONE. After careful consideration, we feel that it has merit but does not fully meet PLOS ONE’s publication criteria as it currently stands. Therefore, we invite you to submit a revised version of the manuscript that addresses the points raised during the review process.

We would appreciate receiving your revised manuscript by Jun 05 2020 11:59PM. To enhance the reproducibility of your results, we recommend that if applicable you deposit your laboratory protocols in protocols.io, where a protocol can be assigned its own identifier (DOI) such that it can be cited independently in the future. For instructions see: http://journals.plos.org/plosone/s/submission-guidelines#loc-laboratory-protocols

We look forward to receiving your revised manuscript.

Kind regards,

Takefumi Nakazawa

Academic Editor

PLOS ONE

Journal Requirements:

2. Thank you for including your ethics statement:  “Animals were collected during IFREMER annual surveys. Protocol was validated by chief scientist of the survey”   

To comply with PLOS ONE submissions requirements, please provide the following information in the Methods section of the manuscript and in the “Ethics Statement” field of the submission form (via “Edit Submission”):  

a.    Please amend your current ethics statement to include the full name of the ethics committee that approved your specific study.

b.    Please amend your current ethics statement to confirm that your named ethics committee specifically approved this study.

For additional information about PLOS ONE submissions requirements for ethics oversight of animal work, please refer to http://journals.plos.org/plosone/s/submission-guidelines#loc-animal-research  

3. In your Methods section, please provide additional location information of the survey sites, including geographic coordinates for the data set if available.

4. To comply with PLOS ONE submissions requirements, please provide methods of sacrifice in the Methods section of your manuscript.

Additional Editor Comments:

I have received the comments from two expert reviewers. Although reviewer #2 is relatively positive, both reviewers raised several critical concerns. (1) For example, reviewer #1 commented that SI data of predators cannot be interpreted appropriately without those of prey or food-web baseline. As a result, the discussion is speculative or not well integrated. This is a common concern that is apparent prior to research. You may address this problem by focusing on the size dependence of feeding habits under the assumption that all size classes have the same baseline (if this is truely reasonable) or even by removing SI data from the manuscript. (2) Also, I agree with Reviewer #1 that you may consider quantitative data on predator body size rather than categorizing them into different size classes. I suggest that re-analyses of the data would be necessary. (3) Reviewer #2 suggested that all the descriptions about optimal foraging theory should be removed, which I totally agree with.

In addition, I found other concerns based on my own reading. Please consider the followings as well.

L147-152: These results are natural because the sample sizes are almost constant across size classes (i.e., the sample sizes do not reflect size distribution).

L158-167: The results of taxon-specific %N are not presented anywhere (the same is true for some parts of Discussion).

L210-: Many parts of Discussion have argued that the present findings are consistent with previous ones. If so, I suggest that such information should be provided in Introduction. Then, please state the novelty of this work more clearly.; i.e., what has been done and remains to be done in this system or other similar systems (e.g., the same species/other species in this/other areas) in Introduction.

L292-: It is not surprising that SI data are not consistent with SC data, as you have supposed in M&M (L109-113). Although you have discussed several hypotheses, another plausible explanation would be that SC data could be more reliable if prey biomass is considered. You may discuss this possibility.

Overall, I suggest that almost all parts of the manuscript should be rewritten. I recognize that revision efforts will be too much, but I basically encourage publications of such descriptive information of behavior and ecology. I hope you to consider all the comments carefully for resubmission.

Reviewers' comments:

Reviewer's Responses to Questions

**Comments to the Author**

1. Is the manuscript technically sound, and do the data support the conclusions?

Reviewer #1: No

Reviewer #2: Yes

2. Has the statistical analysis been performed appropriately and rigorously? 

Reviewer #1: No

Reviewer #2: Yes

3. Have the authors made all data underlying the findings in their manuscript fully available?

Reviewer #1: Yes

Reviewer #2: Yes

4. Is the manuscript presented in an intelligible fashion and written in standard English?

Reviewer #1: Yes

Reviewer #2: Yes

5. Review Comments to the Author

Reviewer #1: This manuscript researches ontogenetic and seasonal diet change of whiting in the Eastern English Channel and Southern North Sea. Diet components of the fish were investigated from stomach contents and stable isotope analysis from 248 and 212 samples, respectively. The relationship between diet (presence or absence of each diet item for SCA and 13C and 15N values for SIA) and explanatory variables (body size classes and season) was assessed by Generalized Linear Models (though it was unclear why they divided their data into 6 age classes without using body size for individual fish). Stomach contents analysis revealed that the frequency of fish increased with fish size, whereas that of pelagic crustaceans decreased vice versa. Although authors claimed that their SIA data somewhat agreed with that of their SCA, they appeared at a loss over how to interpret their results in relation to the SIA, especially for larger fish.

The methods and discussion related to SCA seem elaborate and well thought out. However, the SIA of this study critically lacks important data for proper interpretation of them. Please be aware that SI data for bulk tissue of marine organisms cannot be properly interpreted without understanding of isotope ratios of potential diet and baseline. Trophic position of target fish is nothing more than one of a potential factors to determine their isotope ratios, but baseline shift often results in larger isotopic variation than dietary shift (see Reddin et al. 2017). Specifically, the habitat of studied species is likely to involve high variation in isotopic baseline, due to a large difference in the water depth within the region. Therefore, SI data presented in this study are intrinsically uninterpretable in current form and do not support any results of SCA.

Another issue in SIA of this study is that the authors assumed the same turnover time of muscle among body size classes. In the case of soft tissues like muscle, isotope turnover includes two different factors; metabolic tissue replacement and incremental growth of the tissue. Generally, young fish have high growth rate and active incremental growth of muscle results in their rapid turnover time, whereas old fish are only influenced by metabolic tissue replacement and thus their turnover tend to be slow (see Vander Zanden et al. 2015). In addition, turnover time often differ between 13C and 15N; 13C generally have higher turnover rate than 15N. Given such an inconsistent turnover in muscle among growth stages of whiting, I wander the seasonal comparison of this study is really meaningful.

On the basis of these shortcomings in SIA of this study, I cannot recommend to publish this paper in its current form. I urge you to rewrite the manuscript by taking into account the isotope baseline and variation in turnover time of fish with different growth stage, or completely remove the description related to SIA from the paper.

Reddin, C. J., Bothwell, J. H., O'Connor, N. E., & Harrod, C. (2018). The effects of spatial scale and isoscape on consumer isotopic niche width. Functional Ecology, 32(4), 904-915.

Vander Zanden, M. J., Clayton, M. K., Moody, E. K., Solomon, C. T., & Weidel, B. C. (2015). Stable isotope turnover and half-life in animal tissues: a literature synthesis. PloS one, 10(1).

Detailed comments line by line:

Table 1: Why predicted mean isotope ratios completely agreed with measured values?

Line 120: This is incorrect equation. Add “(‰)” after “δX”.

Line 129: Just using fish size as explanatory variable (not size classes) is better way for this analysis. Using size classes instead of actual size results in degraded levels of measurement of your data.

Lines 235-236: Although stable carbon isotope ratios are conventionally used as pelagic food web tracers, no studies have tested general applicability of such use of 13C. In the case of the North Sea, spatial distribution of 13C values are clearly inconsistent with that of pelagic water (see Trueman et al. 2017).

Lines 258-259: I bet the result reflected migration ability of the fish. Overall, the study site is shallow and will have high 15N value, because denitrification in sediments increase the 15N value of pore water, and phytoplankton intake the 15N-enriched nitric acid. Therefore, smaller fish with low migration ability had higher 15N values. On the other hand, some large fish with high migration ability came from outside the shallow sea area will have lower 15N values. An isoscape of 15N (Trueman et al. 2017) of this area support this explanation.

Reviewer #2: In this paper the authors study the effect of ontogeny and season on the diet of whiting fish. They use two complementary methods that provide quantification over short-term feeding – stomach content analyses, and over longer term, stable isotope analyses, which provide support for ontogenetic diet shift from crustaceans to fish diet. Overall, I think this study provides interesting information on the feeding ecology of this species.

I have a couple general comments:

(i) I would suggest refraining to make any statement that this study’s results are consistent optimal foraging theory. The authors use the idea that, according to OFT, the diet selected should “maximize the energy intake per unit handling time” (l.248). This may be true, but this aspect of whiting’s ecology is not investigated at all in this study. The authors only make the assumption that the observed ontogenetic diet shift can be explained from this statement, but their data do not provide any support to this claim.

(ii) In the current form, the results of the stable isotope analysis are a little hard to interpret for someone with not much background on stable isotope analysis. It would be nice if the authors add a few sentences to explain briefly what general information 13C and 15N content provide about a species’ ecology. Right now the authors mostly present the result of isotope analysis as being “consistent” with stomach content analysis (except for 15N pattern in winter). Is that all this analysis does, confirming the results of stomach content analysis? Or does it also indicate something more?

Other comments:

L171-172: “As whiting grows, it consumes more fish, both in occurrence and abundance.” Looking at figure 4, it indeed seems true during winter, but not during fall, as class 15-20 cm, the second smallest class, has the highest Oi, and Oi remains more or less constant in bigger size class. For abundance, it is individuals between 15-25 cm that feed the most on fish.

L287: typo: “forage”. Do you mean forager? Foraging?

Table 1: I don’t really see the point of the columns on predicted values of δC13 and δC15. The predicted mean values are exactly the same than the observed ones, only the SD changes. Besides, I’m wondering if the SD values reported there may not be in fact estimates of the standard error, as it is the SE usually reported in GLM model analyses… and SD should not change anyway…

Fig 3.A: The maximum size class is 35+, so I am not sure why the graph continues beyond that point… In fig 3B, this is where it stops…

Fig 3. And 4A: It took me some time to figure out that the difference in the data plotted is that Fig 3 (%O) and fig 4A (Oi), make it more explicit in figure’s legends. They are essentially the same data presented differently, so I wonder if Fig. 4A is truly necessary. Also the Y-axes should be labelled

Figure 4: Is the “error bar” SD or SE?

Figure 4: The bar from the 10-15 cm class in winter should be completely removed, as n=0.

Figure 4 – plot of pelagic crustaceans abundance: the value of the 15-20cm in winter is very high making it impossible to see what’s going on for all the other measurements. I think the authors should make one of those graphs with a break on the y-axis, from e.g. 10 to 60 cm.

Figure 4. typo on the lowest-left graph: “Undicentified crustaceans”

Fig. 5 legend should make reference to the color that corresponds to the season.

6. PLOS authors have the option to publish the peer review history of their article (what does this mean?). If published, this will include your full peer review and any attached files.

Reviewer #1: No

Reviewer #2: No

---

## [Author Response · Author response to Decision Letter 0]

11 Jun 2020

Journal Requirements:

1. Thank you for including your ethics statement: “Animals were collected during IFREMER annual surveys. Protocol was validated by chief scientist of the survey” 

To comply with PLOS ONE submissions requirements, please provide the following information in the Methods section of the manuscript and in the “Ethics Statement” field of the submission form (via “Edit Submission”): 

a. Please amend your current ethics statement to include the full name of the ethics committee that approved your specific study.

b. Please amend your current ethics statement to confirm that your named ethics committee specifically approved this study.

The following sentence was included in the material and methods section of the manuscript: 

L. 99-102 (on the manuscript without track changes): « Protocols of all surveys are currently being evaluated by the French research institute for exploitation of the sea (Ifremer) and are validated by the ICES IBTS International Group [41]. In addition, survey's PIs received training about animal well-being and ethics. »

For additional information about PLOS ONE submissions requirements for ethics oversight of animal work, please refer to http://journals.plos.org/plosone/s/submission-guidelines#loc-animal-research

2. In your Methods section, please provide additional location information of the survey sites, including geographic coordinates for the data set if available.

Locations of the sampling stations are represented In Figure 1, with actual coordinates displayed in the supplementary appendix 1B and in the dataset available at the following address: https://doi.org/10.17882/72233

L. 95-96: “Sampling locations are presented in the map (Fig. 1), and actual coordinates are presented in the supporting materials (S1 Fig. and Table)

3. To comply with PLOS ONE submissions requirements, please provide methods of sacrifice in the Methods section of your manuscript.

Organisms were sacrificed following the standard international IBTS protocol, validated by the dedicated ICES international working group. This clarification was included in the material and method section of the manuscript.

L. 110-111: “Organisms were sacrificed following the standard international IBTS protocol [41].”

Additional Editor Comments:

I have received the comments from two expert reviewers. Although reviewer #2 is relatively positive, both reviewers raised several critical concerns. 

1. For example, reviewer #1 commented that SI data of predators cannot be interpreted appropriately without those of prey or food-web baseline. As a result, the discussion is speculative or not well integrated. This is a common concern that is apparent prior to research. You may address this problem by focusing on the size dependence of feeding habits under the assumption that all size classes have the same baseline (if this is truely reasonable) or even by removing SI data from the manuscript. 

The bivalve Aequipecten opercularis (queen scallops) was chosen as baseline, following most classical approach (e.g., Jennings and van der Molen 2015, reference 20 in the plain text). Since A. opercularis and whiting were not always collected in the same stations, a geostatistical interpolation (namely kriging) of the baseline stable isotope ratios were performed in order to consider its spatial variation. Interpolation is based on isotopic ratios, (1) measured on samples collected in the present work simultaneously with fish, (2) measured during previous projects and unpublished yet and, (3) retrieved from literature (Jennings and Warr 2003). The following section detailing the interpolation method was added to the material and methods section of the manuscript. 

L. 168-177: “A station-specific baseline value was used to take into account the spatial variability of isotopic ratios (S1 Fig and Table). As A. opercularis samples could not always be found in the stations where fish were collected, missing δ15N values were calculated with a geostatistical interpolation of baseline δ15N values developed by Kopp et al. [30], and using A. opercularis isotopic ratios obtained in the present and previous published [44] and unpublished sudies (P. Cresson unpubl. data). The dependency of covariance on distance between two sampling sites was modelled using a non-linear regression based on weighted least squares. The weight is the number of pairs of points per distance intervals. Spatial covariance in δ15N was best described by a Gaussian model of distance with adjR2= 0.99.”

Furthermore, the spatial variation of A. opercularis δ15N ratios, whether measured or interpolated, is now presented in a map added in the supplementary appendix, making it possible to estimate a baseline value at each station where whiting were collected. The predicted values are presented in the table of the supplementary table 1 (S1 Table). The inclusion of the baseline allowed us to calculate the trophic level of each individual, i.e., providing an absolute value that takes the spatial variability into account. 

However, since we do not have input data for some coastal sampling stations or some stations with local specificities (e.g. fish farm neighborhood), the accuracy of the values interpolated at these stations may be questioned, leading to a potential overestimation of some trophic level values, particularly for smallest individuals collected in winter. A cautious point was added in the discussion section of the manuscript. 

L. 279 – 284:” Previous studies based on stable isotopes confirmed the important benthic-pelagic coupling in this area, but were not able to identify mechanisms at play, i.e., if coupling results from the consumption of benthic invertebrates by pelagic fish or from the integration of benthic production in the pelagic pathway [22,30]”.

L. 410 – 419: “Unfortunately, no information on isotopic ratios of coastal queen scallops could be collected as the survey trawl could not be operated in shallow waters. Thus, as kriging is directly dependent upon input data, local influences on queen scallops isotopic ratios could not be efficiently captured, and may have induced an overestimation of small whiting δ15N values and trophic levels. Interestingly, estimation of trophic level was associated with highest uncertainties at coastal locations in the model of Jennings and van der Molen [20], as a result of the effect of salinity on baseline δ15N ratios. The high concentration of juveniles in coastal areas, and their limited movement abilities may explain why this influence of terrigenous or fish farm derived nitrogen would primarily affect the smallest size class.”

2. Also, I agree with Reviewer #1 that you may consider quantitative data on predator body size rather than categorizing them into different size classes. I suggest that re-analyses of the data would be necessary. 

GLM analyses were performed with both continuous fish sizes and discrete-size categories. Final analyses considered discrete size classes for several reasons: 

(1) For stomach contents analyses, size-class analyses allow to account for the highly non-linear character of dietary mechanisms. Each individual within a size class is considered as a replicate of the class. Having the diet of some individuals within a size-class can be viewed as a way to capture the average diet of a class, and to limit the importance of extreme feeding mechanisms. This approach is typically used when ontogenetic feeding changes are studied using stomach contents (see by example Graham et al. 2007, Demain et al. 2012, Ross et al. 2016, references 37, 25 and 24 in the plain text).

(2) Using discrete size classes in isotopic analyses GLMs was also justified a posteriori by the non-monotonic character of the isotopic ratios versus size relationships. 

The Table below presents for information the GLM results obtained using continuous fish sizes (not shown in the MS).

Response variable df Residual deviance Explanatory variables df Deviance Residual deviance df p-value

(Chi-square)

Occurrence – fish 245 308.47 Size class 1 17.92 323.04 244 <0.001

 Season 1 5.44 611.30 243 0.02

 Size class × season 1 9.13 587.80 242 0.003

Occurrence -

Benthic crustaceans 245 304.06 Size class 1 6.66 321.51 244 0.01

 Season 1 3.19 318.32 243 0.07

 Size class × season 1 14.27 304.06 242 <0.001

Occurrence -

Pelagic crustaceans 245 225.50 Size class 1 68.33 247.54 244 <0.001

 Season 1 20.32 227.22 243 <0.001

 Size class × season 1 1.72 225.50 242 0.19

Occurrence - 

Unidentified crustaceans 245 303.15 Size class 1 4.77 306.99 244 0.03

 Season 1 2.30 304.69 243 0.13

 Size class × season 1 1.55 303.15 242 0.21

Occurrence - Mollusks 245 196.92 Size class 1 9.85 198.47 244 0.002

 Season 1 1.36 197.11 243 0.24

 Size class × season 1 0.20 196.92 242 0.66

Abundance - fish 245 291.90 Size class 1 25.68 313.18 244 <0.001

 Season 1 12.78 300.40 243 <0.001

 Size class × season 1 8.50 291.90 242 0.004

Abundance -

Benthic crustaceans 245 2069.0 Size class 1 390.59 2146.5 244 <0.001

 Season 1 77.17 2069.4 243 <0.001

 Size class × season 1 0.37 2069.0 242 0.54

Abundance - 

Pelagic crustaceans 245 3411.9 Size class 1 3344.3 7874.0 244 <0.001

 Season 1 4394.2 3479.7 243 <0.001

 Size class × season 1 67.8 3411.9 242 <0.001

Abundance -

Unidentified crustaceans 245 587.80 Size class 1 68.47 683.57 244 <0.001

 Season 1 72.28 611.30 243 <0.001

 Size class × season 1 23.50 587.80 242 <0.001

Abundance - Mollusks 245 254.58 Size class 1 37.96 256.02 244 <0.001

 Season 1 0.05 255.97 243 0.82

 Size class × season 1 1.38 254.58 242 0.24

δ13C 211 70.77 Size class 1 1.63 76.77 210 0.03

 Season 1 3.52 73.25 209 0.002

 Size class × season 1 2.48 70.77 208 0.007

δ15N 211 160.89 Size class 1 1.45 273.88 210 0.17

 Season 1 49.37 224.52 209 <0.001

 Size class × season 1 63.63 160.89 208 <0.001

Trophic level 211 13.92 Size class 1 0.008 22.41 210 0.75

 Season 1 3.99 18.42 209 <0.001

 Size class × season 1 3.09 15.34 208 <0.001

3. Reviewer #2 suggested that all the descriptions about optimal foraging theory should be removed, which I totally agree with.

We have removed all references to the optimal foraging theory in the manuscript. We replaced these assumptions by Werner and Gilliam's hypothesis (1984) related to the trade-off between foraging and predation risk. In the introduction section of the manuscript, we have rewritten the part related to ontogenetic dietary changes. In the rest of the manuscript, all parts concerning the optimal foraging theory have been deleted.

L. 33 – 40: “Organism’s energetic requirements differ according to their size. Ontogenetic dietary shifts, the changes in resource use over the lifespan of a consumer, are widespread for many taxa [10,11]. Consumers make trade-offs between foraging and mortality risk [10]. These trade-off changes according to ontogeny depending on the balance between the risk of mortality due to predation and the benefits received from the resource. Several biotic (predation risk, competition, prey availability) or abiotic (habitat use) factors may be responsible of these variations [12]. In addition, species intrinsic factors, such as the gape size or swimming abilities promote these ontogenetic changes.”

In addition, I found other concerns based on my own reading. Please consider the followings as well.

4. L147-152: These results are natural because the sample sizes are almost constant across size classes (i.e., the sample sizes do not reflect size distribution).

We agree with the reviewer, and this was actually a purpose of this sampling. The sampling was not designed to represent the natural variability of the size distribution, but to capture the size-driven variability in diet. So, as stomach content analysis requires more samples than stable isotope analysis to statistically examine differences among size class (see by example Hyslop 1980, Cresson et al. 2014, references 43 and 38 in the manuscript), the sampling was refined. The representativeness of the subsample was confirmed by the similarity in average size between the whole dataset and the subsample used for stable isotope analysis. A sentence detailing this point was added:

L. 203-204: “Subsample used for stable isotopes analyses can thus be considered representative of the larger sample used for stomach content analyses.”

5. L158-167: The results of taxon-specific %N are not presented anywhere (the same is true for some parts of Discussion).

A figure representing %N was added in the supplementary appendix of the manuscript (S2 Fig.). For each individual the taxon-specific results are presented in the dataset presented at the following address: https://doi.org/10.17882/72233

L. 218 – 219: “The taxon-specific results can be found in the dataset published at the following address: https://doi.org/10.17882/72233.”

6. L210-: Many parts of Discussion have argued that the present findings are consistent with previous ones. If so, I suggest that such information should be provided in Introduction. Then, please state the novelty of this work more clearly.; i.e., what has been done and remains to be done in this system or other similar systems (e.g., the same species/other species in this/other areas) in Introduction.

The trophic ecology of whiting has been studied in several ecosystems. However, all biotic and abiotic conditions between ecosystems can lead to differences in feeding habits. In the EEC-SNS, the trophic ecology of whiting was mainly studied by stomach contents analyses. A Previous study using stable isotope analyses have not investigated the extent to which it was affected by ontogenetic and seasonal changes (see by example Kopp et al. 2015, reference 30 in plain text). In this study, we propose to study whiting trophic ecology using a combination of isotopic and stomach content analyses to provide information at several time scales. Our sampling allows us to study ontogenetic variations and to consider seasonal differences of whiting diets. We have added a paragraph in the introduction section of the manuscript on this subject: 

L. 56 – 74: “The diet of this high trophic level species is well investigated in many ecosystems because it predates on commercially exploited fish and invertebrates, and also since it can compete with other economically valuable gadoid species [24]. Most studies highlight the main consumption of fish and invertebrates (e.g. crustaceans, mollusks or annelids). Some of them report an ontogenetic change in whiting diet, with small individuals mostly consuming crustaceans, and larger whiting mostly consuming fish. This feeding pattern has been previously observed in other ecosystems, i.e., the North Sea [24–27], Baltic Sea [24], Irish Sea [28] and Celtic Sea [27,29]. However, ecosystems are subject to different environmental and ecological conditions. All differences in prey availability, predator abundance, fishing pressure and abiotic features make differences between the ecological role and diet of M. merlangus in an ecosystem and its conspecific in other ecosystems. To our knowledge, studies that have investigated EEC whiting diets have not focused on ontogenetic and seasonal diet changes [30,31]. The increased trophic importance of fish with ontogeny is consistent with outputs of an independent modeling study based on an EEC application of the OSMOSE ecosystemic model [15]. Except this modeling study, the trophic ecology of whiting in EEC-SNS was mainly studied by stomach content analyses, providing a short-term view of its diet [24,25,31]. The use of stable isotope analyses in addition to analyses of stomach contents helps to resolve some common biases related to the analysis of stomach contents.”

7. L292-: It is not surprising that SI data are not consistent with SC data, as you have supposed in M&M (L109-113). Although you have discussed several hypotheses, another plausible explanation would be that SC data could be more reliable if prey biomass is considered. You may discuss this possibility.

We recognize that consideration of prey biomass would enable a better understanding of trophic ecology, and would allow the integration of additional indices such as the index of relative importance taking into account both abundance, occurrence, and prey biomass. However, although fish were immediately frozen after capture to limit digestion, many preys were too degraded to estimate their biomass accurately. This may question the relevance of using biomass as a reliable descriptor of fish diet (Baker et al. 2014). Biomass of remaining hard pieces (crustacean claws, fish otoliths etc.) cannot be easily converted into the actual biomass of the original prey consumed without accurate mass-mass or length-mass relationships. Including biomass as a descriptor of fish diet would also increase the well-known bias due do different prey digestibility, since the ratio between actual mass (i.e. the requested information) and measured mass of prey at different stages of degradation would not be similar,

For this reason, we only considered in our analyses prey occurrence and abundance.

Overall, I suggest that almost all parts of the manuscript should be rewritten. I recognize that revision efforts will be too much, but I basically encourage publications of such descriptive information of behavior and ecology. I hope you to consider all the comments carefully for resubmission.

Reviewer #1’s comments

This manuscript researches ontogenetic and seasonal diet change of whiting in the Eastern English Channel and Southern North Sea. Diet components of the fish were investigated from stomach contents and stable isotope analysis from 248 and 212 samples, respectively. The relationship between diet (presence or absence of each diet item for SCA and 13C and 15N values for SIA) and explanatory variables (body size classes and season) was assessed by Generalized Linear Models (though it was unclear why they divided their data into 6 age classes without using body size for individual fish). Stomach contents analysis revealed that the frequency of fish increased with fish size, whereas that of pelagic crustaceans decreased vice versa. Although authors claimed that their SIA data somewhat agreed with that of their SCA, they appeared at a loss over how to interpret their results in relation to the SIA, especially for larger fish.

The methods and discussion related to SCA seem elaborate and well thought out. 

1. However, the SIA of this study critically lacks important data for proper interpretation of them. Please be aware that SI data for bulk tissue of marine organisms cannot be properly interpreted without understanding of isotope ratios of potential diet and baseline. Trophic position of target fish is nothing more than one of a potential factors to determine their isotope ratios, but baseline shift often results in larger isotopic variation than dietary shift (see Reddin et al. 2017). Specifically, the habitat of studied species is likely to involve high variation in isotopic baseline, due to a large difference in the water depth within the region. Therefore, SI data presented in this study are intrinsically uninterpretable in current form and do not support any results of SCA.

The point you raise has also been mentioned by the editor. You will find an answer to this point in the first comment from the editor (see above).

2. Another issue in SIA of this study is that the authors assumed the same turnover time of muscle among body size classes. In the case of soft tissues like muscle, isotope turnover includes two different factors; metabolic tissue replacement and incremental growth of the tissue. Generally, young fish have high growth rate and active incremental growth of muscle results in their rapid turnover time, whereas old fish are only influenced by metabolic tissue replacement and thus their turnover tend to be slow (see Vander Zanden et al. 2015). In addition, turnover time often differ between 13C and 15N; 13C generally have higher turnover rate than 15N. Given such an inconsistent turnover in muscle among growth stages of whiting, I wander the seasonal comparison of this study is really meaningful. 

On the basis of these shortcomings in SIA of this study, I cannot recommend to publish this paper in its current form. I urge you to rewrite the manuscript by taking into account the isotope baseline and variation in turnover time of fish with different growth stage, or completely remove the description related to SIA from the paper.

Turnover rate is often expressed as isotopic half-life, defined as the time required reaching 50% equilibration with diet. Vander Zanden et al (2015) have developed equations to estimate the turnover rate, which varies according to the type of organism (endotherm vs. ectotherm) and the type of tissue. For fish muscle, the equation is as follows: 

ln(half life) = 0.21 . ln( body mass) + 3.23 

In our study, whiting belonging to the smallest size class have an average mass of 15g. Those in the largest size class have an average mass of 508g. By applying the above equation, we can then estimate a turnover of about 45 days for the smallest individuals and 93 days for the largest, i.e., about three months. Considering a turnover rate of several months for adults is classical for isotope analyses (see for example Hesslein et al., 1993, McIntyre & Flecker, 2006, Nielsen et al., 2018, references 86, 87 and 82 in the plain text). 

We then admit that juveniles’ isotope ratios may reflect feeding during the collection season (autumn or winter). For adults, since the turnover time is estimated to be about three months, this may correspond to feeding prior to the collection season.

In the discussion, we added a section on the potential differences of turnover between juveniles and adults. We thank the reviewer for making us think about this possibility. 

L. 418 – 446 : “Stomach content analysis provides short-term information about the integrated diet, from hours to days [43,82]. Stable isotopes analysis provides information on assimilated diet at a longer time scale than stomach contents, but varies according to the lifecycle of species. A meta-analysis carried out on several taxa revealed that, in general, turnover rate differs between juveniles and adults. Stable isotopes ratios change due to growth and metabolic replacement associated with a change in diet were observed between juveniles and adults [83]. These authors have developed an equation to estimate the turnover rate according to the body mass (proxy of the size) and to the type of species (endotherm vs. ectotherm) and tissue. Tissues characterized by a quick turnover provide the most recent dietary information. Using this equation, the turnover rate was estimate to ~45 days for juveniles (15 g average mass) which is consistent with the turnover rate estimated for cod, another gadoid species [84]. We then admit that juveniles ‘isotope ratios may reflect feeding during the collection season (autumn or winter). For adults (weighing 508 g on average), the turnover rate was of about three months, consistently with the expected turnover of adult fish [82,85,86]. Measured isotopic ratios may partly result from dietary elements integrated in the tissues before the sampling season, and away from EEC, due to movement abilities of adult whiting [87]. However, isotopic integration of diet has to be viewed from a dynamic point of view. Addressing the seasonal (i.e. in autumn and winter) and spatial (i.e. over the potential area covered by whiting over a three months period) variability of prey isotopic ratios would also be needed. Finally, since the integration of isotopes in tissues is dynamic, additional studies are necessary to know precisely the rate at which isotopes are incorporated into whiting’s muscle and the discrimination factor between tissues and diet [88]. It should also be noted that the method used to estimate turnover rates does not distinguish between isotopes (δ13C, δ15N). However, several studies have shown that for fish, the turnover rate of δ13C is faster than that of δ15N due to different catabolic processes operating on the various biochemical constituents of tissues (amino acids, proteins, lipids [89–91]). Consequently, both δ13C and δ15N for the same species inform about diet at different time scales, highlighting the importance of considering the turnover rate to know when that diet has been assimilated for a better understanding of species diet.”

Detailed comments line by line:

3. Table 1: Why predicted mean isotope ratios completely agreed with measured values?

This pattern results from the fact that Isotopic ratios were analyzed using a standard linear model with a Gaussian error structure. This would not necessarily be true with Generalized Linear Models and using alternative error types. We illustrate below how predictions were calculated.

Considering that the prediction (Y) = A + Bi + Cj + Di,j with

A= intercept, Bi = estimate for the size class i + Cj = the estimate of the season j and Di,j the interaction between size class i and season j. 

Below is an example for the predicted �15N values

Prediction for the 20-25cm size class in winter: 

Y = 14.4194 + 1.6028 + 0.4151 + 0.2250 = 16.66

We removed predicted mean isotope ratios in the manuscript 

4. Line 120: This is incorrect equation. Add “(‰)” after “δX”.

Done 

5. Line 129: Just using fish size as explanatory variable (not size classes) is better way for this analysis. Using size classes instead of actual size results in degraded levels of measurement of your data.

This point has also been mentioned by the editor. See the answer to this point in the second comment from the editor (see above).

6. Lines 235-236: Although stable carbon isotope ratios are conventionally used as pelagic food web tracers, no studies have tested general applicability of such use of 13C. In the case of the North Sea, spatial distribution of 13C values are clearly inconsistent with that of pelagic water (see Trueman et al. 2017).

We agree with the reviewer: due to the relative nature of stable isotopes, an isotopic ratio per se cannot be considered as tracer of pelagic or benthic food web. 

Nevertheless, several papers, in the EEC and other ecosystems, have demonstrated that the comparison of isotopic ratios can be used to track the relative importance of benthic or pelagic pathways, notably as carbon isotopic ratios are largely different between benthic and pelagic pathways (~2 to 3 ‰ in the EEC; Kopp et al. 2015; Cresson et al. in press) and are largely higher than the spatial variability. Increasing importance of benthic sources results in higher (less negative) isotopic ratios. In the above-mentioned sentence, we interpreted the increase in isotopic ratios with size from a relative point of view, i.e., assuming an increased contribution of benthic pathway in the diet of larger whiting individuals. We understand that this assumption was not clear enough. The sentence was thus modified in the MS as follows:

L. 290 - 292: “Since pelagic organisms are generally δ13C-depleted compared to benthic ones [30,35] the increase of �13C values is thus consistent with a diet switch from pelagic invertebrates to benthic fish as whiting grows.”

7. Lines 258-259: I bet the result reflected migration ability of the fish. Overall, the study site is shallow and will have high 15N value, because denitrification in sediments increase the 15N value of pore water, and phytoplankton intake the 15N-enriched nitric acid. Therefore, smaller fish with low migration ability had higher 15N values. On the other hand, some large fish with high migration ability came from outside the shallow sea area will have lower 15N values. An isoscape of 15N (Trueman et al. 2017) of this area support this explanation.

We thank the reviewer for this hypothesis. We have added it to the discussion

L. 312– 326: “The study site is shallow and presumably has high δ15N value, because denitrification in sediments increase the δ15N value of pore water, and consequently of the organic matter at the base of food webs [64,65]. Therefore, smaller fish with low migration ability had higher δ15N values. On the other hand, since isotopic ratios of the baseline vary spatially [44,48], we can hypothesize that the decrease in δ15N for larger individuals does not reflect a different diet, but an artifact related to a spatially different baseline. As mentioned by Reddin et al. [49], from an isotopic point of view, baseline can sometimes be more variable than the diet. Here, the difference between the minimal and maximal δ15N values kriged for queen scallops was of ~1.5 ‰ (S1 Table). However variations may be more important if considering neighboring areas, i.e. the Western English Channel or northern areas in the North Sea [44,48]. Assuming high migration abilities for larger whiting, their isotopic ratios may reflect a diet in areas where baseline δ15N is markedly lower. Investigating isotopic ratios of whiting over a large spatial continuum, with samples from the whole English Channel and the North Sea could thus be useful to better understand the influence of spatial migration on fish isotopic ratios.”

Reviewer #2’s comments

In this paper the authors study the effect of ontogeny and season on the diet of whiting fish. They use two complementary methods that provide quantification over short-term feeding – stomach content analyses, and over longer term, stable isotope analyses, which provide support for ontogenetic diet shift from crustaceans to fish diet. Overall, I think this study provides interesting information on the feeding ecology of this species.

I have a couple general comments:

1. I would suggest refraining to make any statement that this study’s results are consistent optimal foraging theory. The authors use the idea that, according to OFT, the diet selected should “maximize the energy intake per unit handling time” (l.248). This may be true, but this aspect of whiting’s ecology is not investigated at all in this study. The authors only make the assumption that the observed ontogenetic diet shift can be explained from this statement, but their data do not provide any support to this claim.

This point has also been mentioned by the editor. See the answer to this point in the third comment from the editor (see above).

2. In the current form, the results of the stable isotope analysis are a little hard to interpret for someone with not much background on stable isotope analysis. It would be nice if the authors add a few sentences to explain briefly what general information 13C and 15N content provide about a species’ ecology. Right now the authors mostly present the result of isotope analysis as being “consistent” with stomach content analysis (except for 15N pattern in winter). Is that all this analysis does, confirming the results of stomach content analysis? Or does it also indicate something more?

Although explanations of the δ13C and δ15N values were present in the material and methods section of the previous version, as well as the complementary vision between isotopes and stomach content analyses, these explanations may not have been sufficiently clear or emphasized. We then added a new paragraph on this subject in the introduction section of the revised version of the manuscript .Indeed, isotope analyses do not only confirm or not the results of stomach contents analyses, but also provide additional information at different time scales. We thank the reviewer for pointing this. 

L. 75 - 85: “When consuming a prey, a predator integrates the carbon and nitrogen isotopic ratios of its prey into its own tissues, with a difference called isotopic discrimination [32]. The nitrogen isotopic ratio (δ15N), theoretically enriched by ~3.4 ‰ per trophic level, is classically used as an estimator of the trophic position [33]. Carbon isotopic ratio (δ13C) is less enriched (~1 ‰ per trophic level), allowing its use as tracer of the origin of food sources [34]. In the marine environment, δ13C is used to distinguish between benthic and pelagic sources because benthic producers are typically 13C-enriched relative to pelagic producers [35]. Unlike stomach content analyses, stable isotope ratios of an organism’s tissues provide information on the time-integrated assimilated diet. Combining stable isotopes and stomach content analyzes is thus recognized as a powerful approach to gain accurate and complementary vision of fish trophic patterns at different time scales [36–38].”

In addition, as raised by reviewer 1 (second comment), differences of turnover rates between juveniles and adults can provide information on diet at different time scales. See above, the second response to reviewer 1 for more explanation about this point.

Other comments:

3. L171-172: “As whiting grows, it consumes more fish, both in occurrence and abundance.” Looking at figure 4, it indeed seems true during winter, but not during fall, as class 15-20 cm, the second smallest class, has the highest Oi, and Oi remains more or less constant in bigger size class. For abundance, it is individuals between 15-25 cm that feed the most on fish.

A clearer distinction was achieved between autumn and winter. Indeed, the increase in fish consumption, both in abundance and occurrence, mainly concerns the winter season. In autumn, it was mainly an increase of the mollusks consumption.

4. L287: typo: “forage”. Do you mean forager? Foraging?

The term « forage » was removed

5. Table 1: I don’t really see the point of the columns on predicted values of δC13 and δC15. The predicted mean values are exactly the same than the observed ones, only the SD changes. Besides, I’m wondering if the SD values reported there may not be in fact estimates of the standard error, as it is the SE usually reported in GLM model analyses… and SD should not change anyway…

The same remark was made by the reviewer 1. An explanation of the predicted values can be found above (third comment of the reviewer 1). Since the predicted values are provided from glm, the term « SD » has been replaced by "SE", more appropriate for model outputs.

6. Fig 3.A: The maximum size class is 35+, so I am not sure why the graph continues beyond that point… In fig 3B, this is where it stops…

This was just a graphical concern on the x-axis. The maximum size class is 35+, and that is where the graph stops, as in Figure 3B. This problem has been solved.

7. Fig 3. And 4A: It took me some time to figure out that the difference in the data plotted is that Fig 3 (%O) and fig 4A (Oi), make it more explicit in figure’s legends. They are essentially the same data presented differently, so I wonder if Fig. 4A is truly necessary. Also the Y-axes should be labelled

In order to avoid redundancies between figure 3 and figure 4, we have moved figure 3 to supplementary appendix (S2 Fig).

Figure 4a shows the probability of occurrence for each size class. Results provided from GLM models. Y-axes were respectively relabeled « Probability of occurrence » and « Number of prey items (Ni) respectively » 

8. Figure 4: Is the “error bar” SD or SE

For Figure 4, since data presented are the result of GLM models, error bars corresponds to SE. This information was added in the legend of the figure

9. Figure 4: The bar from the 10-15 cm class in winter should be completely removed, as n=0.

Done

10. Figure 4 – plot of pelagic crustaceans abundance: the value of the 15-20cm in winter is very high making it impossible to see what’s going on for all the other measurements. I think the authors should make one of those graphs with a break on the y-axis, from e.g. 10 to 60 cm.

Done

11. Figure 4. typo on the lowest-left graph: “Undicentified crustaceans”

The typo was corrected on the graph

12. Fig. 5 legend should make reference to the color that corresponds to the season.

Done

---

## [Decision Letter · Decision Letter 1]

23 Jul 2020

PONE-D-20-06925R1

Seasonal and ontogenetic variation of whiting diet in the Eastern English Channel and the Southern North Sea

PLOS ONE

Dear Dr. Timmerman,

Thank you for submitting your manuscript to PLOS ONE. After careful consideration, we feel that it has merit but does not fully meet PLOS ONE’s publication criteria as it currently stands. Therefore, we invite you to submit a revised version of the manuscript that addresses the points raised during the review process.

We look forward to receiving your revised manuscript.

Kind regards,

Takefumi Nakazawa

Academic Editor

PLOS ONE

Additional Editor Comments (if provided):

I have received the comments from the same reviewers. Both of them appreciated your revision efforts, but they addressed several issues. I hope you to consider all the comments carefully for resubmission.

In addition, I found other concerns based on my own reading. Please consider the followings as well.

I think you misundertood my comments regarding prey biomass.  I did not request you to assess it as an index of fish trophic status. I understand that it is technically difficult as you replied.  What I requested was to discuss the possibility that the consideration of prey biomass may help to solve the gap between stomach content and stable isotope data, even though it is technically difficult, as with other hypotheses you provided.  You replied why it is technically difficult.  That would be by itself a perspective discussion, if it makes sense.

Other specific comments

L35: "trade-offs between foraging and mortality risk"

This can be misleading. The trade-off would occur between the relative benefits (foraging benefits relative to associated costs including mortality risk) of candidate resources, but not between foraging gains and associated cosits.

L319: "smaller fish with low migration ability had higher δ15N values."

Are there any results or references to support this statement?

L339-340: In the EEC-SNS, winter nutrient concentration and temperatures were lower than in autumn.

Are there any results or references to support this statement?

L354-355: Previous studies reported a particularly high energetic value for these species.

Please cite the papers.

Throughout Disucssion, I found several sentences not supported by any results or references.  Please provide supporting evidence for such assertive statements.

Reviewers' comments:

Reviewer's Responses to Questions

**Comments to the Author**

1. If the authors have adequately addressed your comments raised in a previous round of review and you feel that this manuscript is now acceptable for publication, you may indicate that here to bypass the “Comments to the Author” section, enter your conflict of interest statement in the “Confidential to Editor” section, and submit your "Accept" recommendation.

Reviewer #1: All comments have been addressed

Reviewer #2: All comments have been addressed

2. Is the manuscript technically sound, and do the data support the conclusions?

Reviewer #1: Yes

Reviewer #2: Yes

3. Has the statistical analysis been performed appropriately and rigorously? 

Reviewer #1: Yes

Reviewer #2: Yes

4. Have the authors made all data underlying the findings in their manuscript fully available?

Reviewer #1: Yes

Reviewer #2: Yes

5. Is the manuscript presented in an intelligible fashion and written in standard English?

Reviewer #1: Yes

Reviewer #2: Yes

6. Review Comments to the Author

Reviewer #1: In this revision, the quality of this manuscript has greatly improved. Authors appropriately provided isotopic baseline in their study site and (fortunately) revealed that there were no significant variation in isoscape there. Overall, their revisions were satisfactory for me and I recommend to publish this paper once the other minor issues I listed below has been addressed.

Line 14-15: Please add a description of data collection for baseline organism.

Line 77: I suggest to replace “theoretically” with “empirically”.

Line 78-79: This statement is not exactly true because some tissues like bone collagen enrich d13C greater than 1 %.

Line 120, “low lipid content”: This is ok but did you extract lipid from muscle sample? I couldn’t find the description about lipid extraction from the text. Lipid extraction is essential pretreatment for stable isotope analysis of muscle because adipose tissue generally has substantially decreased 13C values compared with muscle, and slight individual difference in the quantity of body fat can result in the difference of 13C.

Lines 152-153: Please include manufacture’s name and location for specific equipment.

Line 162-164: This sentence should be moved to “Studied area and sampling”. Furthermore, detailed information about bivalve sample (e.g. sample size mean and SD of isotope ratios) should be included.

Line 186: I understood why you used 5 size class instead of quantitative body size data. Anyway, it may be helpful for readers to include some justifications about the use of size class in the text.

Line 266, “… from -17.34 ± 0.68‰ (35+) to -16.77 ± 0.68‰ and -16.77 ± 0.65‰ (20-30cm) in winter.”: This sentence is confusing.

Lines 294-296: Again, did you extract lipids from whiting samples? If not, the interpretation of their 13C value is difficult because the difference of lipid concentration may explain the ontogenetic and seasonal difference of 13C values.

Fig. 4. Legends of 13C and 15N are reversed.

Reviewer #2: The authors correctly addressed previous reviewers’ comments, and produced a sound manuscript that is acceptable for publication in PLOS one.

Below, I note a few errors:

Figure 3 is mislabeled: The left panel is 13C, whereas the central panel is 15N

L. 198: change “analyzes” into analyses

L267: change “increases” into increased

7. PLOS authors have the option to publish the peer review history of their article (what does this mean?). If published, this will include your full peer review and any attached files.

Reviewer #1: No

Reviewer #2: No

---

## [Author Response · Author response to Decision Letter 1]

3 Sep 2020

Additional Editor Comments (if provided):

I have received the comments from the same reviewers. Both of them appreciated your revision efforts, but they addressed several issues. I hope you to consider all the comments carefully for resubmission.

In addition, I found other concerns based on my own reading. Please consider the followings as well.

1. I think you misundertood my comments regarding prey biomass. I did not request you to assess it as an index of fish trophic status. I understand that it is technically difficult as you replied. What I requested was to discuss the possibility that the consideration of prey biomass may help to solve the gap between stomach content and stable isotope data, even though it is technically difficult, as with other hypotheses you provided. You replied why it is technically difficult. That would be by itself a perspective discussion, if it makes sense.

The question of the most efficient indices to apply to stomach content analyses for an accurate representation of the diet has been largely debated, is the subject of several meta-analyses and position papers (e.g. Hyslop 1980; Saika 2016; Amundsen and Sánchez‐Hernández, 2019; da Silveira et al., 2020). This question is still discussed, notably in the context of the implementation of the commercial species diet monitoring, to provide the data needed to move towards an ecosystemic management of fisheries. Nevertheless, even if discussions remain, most papers highlight that biomass-based indices are the most biased, and that considering abundance or occurrence is most cautious. We chose to follow this approach in our manuscript. 

For example, biomass of remaining hard pieces (crustacean claws, fish otoliths etc.) cannot be easily converted into the actual biomass of the original prey consumed without accurate mass-mass or length-mass relationships. Including biomass as a descriptor of fish diet would also increase the well-known bias due do different prey digestibility, since the ratio between actual mass (i.e. the requested information) and measured mass of prey at different stages of degradation would not be similar. In other words, assessing the occurrence or abundance of preys is a testimony of what was actually consumed, notably as prey abundance is based on a parsimonious method that considers the lowest possible number of preys that originated the number of remains observed. But the weight of the items remaining in the stomach is the weight of the part not integrated in the predators’ tissues, while stable isotopes provide integrated information about the diet. Consideration of biomass may not necessarily resolve the observed gap between stomach contents and stable isotopes. For this reason, we have not included the consideration of biomass in perspective discussion. 

We assume from editor’s repeated comment that this point was unclear in the manuscript. We thus reorganize the final part of the materials and methods section about stomach content to add a rationale for the use of abundance and occurrence rather than biomass. 

L. 150-161 (on the manuscript without track changes): Abundance and occurrence-based indices were used as they are considered less biased than indices calculated from prey biomass measurement [12,45]. Assessing the occurrence or abundance of preys is a testimony of what was actually consumed, notably as prey abundance is based on a parsimonious method that limits overestimation of the number of degraded items, by estimating the most plausible number of prey individual that have originated the number of remains: paired items (e.g. otoliths, eyes, claws) were counted and divided by two. Other uncountable remains (e.g. muscles, gills, annelids bristles) were not counted and considered as remains of a single individual. On the contrary, biomass based indices are questioned, as they are more affect by well-known biases of stomach content. Differential digestibility of the preys leads to an overestimation of the mass of hard parts, and as biomass of parts found of the stomachs is the biomass of the undigested part of the diet, while stable isotopes reflect the part of the diet integrated in the tissues. Since stomach content analysis only provides a snapshot of fish diet, this approach was combined with stable isotopes analysis of fish tissue, which provides information on the time-integrated assimilated diet [35–37].

Other specific comments

2. L35: "trade-offs between foraging and mortality risk"

This can be misleading. The trade-off would occur between the relative benefits (foraging benefits relative to associated costs including mortality risk) of candidate resources, but not between foraging gains and associated cosits.

The sentence has been modified for a better understanding: 

L. 36-37 (on the manuscript without track changes): “Consumers make trade-offs between the benefits (e.g., foraging) relative to associated costs including mortality”

3. L319: "smaller fish with low migration ability had higher δ15N values."

Are there any results or references to support this statement?

A reference was added to support the fact that juveniles had lower migrations abilities than adults (Cohen 1990, reference number 66 in the plain text). In addition, the sentence was slightly modified in the text: 

L. 335-336 (on the manuscript without track changes): “because whiting migrate only after the first year of life [66], smaller fish with low migration ability had higher δ15N values.” 

4. L339-340: In the EEC-SNS, winter nutrient concentration and temperatures were lower than in autumn.

Are there any results or references to support this statement?

EEC-SNS, temperature and nutrient concentration vary seasonally. Temperatures varied from 4°C in winter to 19°C in summer. In autumn, temperatures begin to decrease but remain higher than in winter. Similarly, in EEC-SNS, a phytoplankton bloom occurs in autumn. This bloom is less important than that of the spring period, but remains higher than in winter. Three references were added to support this statement (Breton et al. 2000, Carpentier et al. 2009, Morris et al. 2018, references 68, 69 and 70 in the plain text). 

5. L354-355: Previous studies reported a particularly high energetic value for these species.

Please cite the papers.

Two references were added to support this statement (Spitz et al. 2010, Cresson et al. 2020, references 63 and 76 in the plain text) 

6. Throughout Disucssion, I found several sentences not supported by any results or references. Please provide supporting evidence for such assertive statements.

Some missing references to support our statements were added throughout discussion

- L. 297-298 (on the manuscript without track changes): “Most studies highlighted the consumption of crustaceans but did not distinguish between benthic or pelagic species [24,26,27] 

- L. 317-318 (on the manuscript without track changes): “The positive relationship between δ15N values and body size is commonly observed for piscivores [13]»

- L. 335-336 (on the manuscript without track changes): “because whiting migrate only after the first year of life [66], smaller fish with low migration ability had higher δ15N values”

- L. 343 (on the manuscript without track changes): “Assuming high migration abilities for larger whiting [66]”

- L. 371-372 (on the manuscript without track changes): “Previous studies reported a particularly high energetic value for these species [63,76]”

- L. 396-397 (on the manuscript without track changes): One of the well-known limitations of stomach contents is the impossibility of identifying soft-bodied prey, such as fish eggs, due to their rapid digestion [44,79]

Reviewers' comments:

Reviewer's Responses to Questions

Comments to the Author

1. If the authors have adequately addressed your comments raised in a previous round of review and you feel that this manuscript is now acceptable for publication, you may indicate that here to bypass the “Comments to the Author” section, enter your conflict of interest statement in the “Confidential to Editor” section, and submit your "Accept" recommendation.

Reviewer #1: All comments have been addressed

Reviewer #2: All comments have been addressed

2. Is the manuscript technically sound, and do the data support the conclusions?

Reviewer #1: Yes

Reviewer #2: Yes

3. Has the statistical analysis been performed appropriately and rigorously? 

Reviewer #1: Yes

Reviewer #2: Yes

4. Have the authors made all data underlying the findings in their manuscript fully available?

Reviewer #1: Yes

Reviewer #2: Yes

5. Is the manuscript presented in an intelligible fashion and written in standard English?

Reviewer #1: Yes

Reviewer #2: Yes

6. Review Comments to the Author

Reviewer #1: In this revision, the quality of this manuscript has greatly improved. Authors appropriately provided isotopic baseline in their study site and (fortunately) revealed that there were no significant variation in isoscape there. Overall, their revisions were satisfactory for me and I recommend to publish this paper once the other minor issues I listed below has been addressed.

1. Line 14-15: Please add a description of data collection for baseline organism.

The following sentence was added: 

L. 15-16 (on the manuscript without track changes): “In addition, queen scallops (Aequipecten opercularis) samples were collected along with fish to be used as isotopic benthic baseline.”

2. Line 77: I suggest to replace “theoretically” with “empirically”.

Done

3. Line 78-79: This statement is not exactly true because some tissues like bone collagen enrich d13C greater than 1 %.

We recognize that for some tissues, the isotope discrimination may be lower or higher than 3.4 and 1‰. We have modified our sentence to focus on muscle, the tissue considered in our study.

L. 79-82 (on the manuscript without track changes): « The muscle nitrogen isotopic ratio (δ15N), empirically enriched by ~3.4 ‰ per trophic level, is classically used as an estimator of the trophic position [32]. Muscle carbon isotopic ratio (δ13C) is less enriched (~1 ‰ per trophic level), allowing its use as tracer of the origin of food sources [33].”

4. Line 120, “low lipid content”: This is ok but did you extract lipid from muscle sample? I couldn’t find the description about lipid extraction from the text. Lipid extraction is essential pretreatment for stable isotope analysis of muscle because adipose tissue generally has substantially decreased 13C values compared with muscle, and slight individual difference in the quantity of body fat can result in the difference of 13C.

Lipid content in fish muscle is generally low, but varies between fish species. The C:N ratio is used as an indicator of the lipid concentration in a sample. When lipid content is low, below 5% lipid (C:N<3.5), lipid correction – whether chemical extraction or mathematical correction of the isotopic ratios – is not requested (Post et al. 2007). In our dataset, C:N ratios ranged between 3.08 and 3.43 (average 3.19 ± 0.08), we did not performed lipid extraction pretreatment. The following sentence was added in the plain text:

L 122-126 (on the manuscript without track changes): White dorsal muscle is the most classical tissue used for stable isotopes analysis in fish because of its low turnover rate and low lipid content [43], notably for whiting, one of the species with lowest C:N ratios and consequently lipid content in the area [44]. As C:N ratios measured in all samples were low (average value: 3.19 ± 0.08; min: 3.08, max: 3.43), raw isotopic ratios were used. 

5. Lines 152-153: Please include manufacture’s name and location for specific equipment.

Manufacture’s name and location were added for specific equipment

“Isotopic analyzes were carried out with a Thermo Fisher Scientific Delta V Plus mass spectrometer (Bremen, Germany), coupled to a Flash 2000 elemental analyzer with a Conflow IV interface (Thermo Scientific, Milan, Italy)” .

6. Line 162-164: This sentence should be moved to “Studied area and sampling”. Furthermore, detailed information about bivalve sample (e.g. sample size mean and SD of isotope ratios) should be included.

The following sentence was moved to the “Studied area and sampling” section. 

L. 111-113 (on the manuscript without track changes): “Queen scallops (A. opercularis) samples were collected and stored frozen to be used as isotopic benthic baseline in the calculation of trophic level, following most classical approach [20,41,42]. “

Furthermore, additional information about bivalve sample was included in the material and method section of the manuscript: 

L. 186-187 (on the manuscript without track changes): “A total of 76 isotopic measurements were used as a baseline for geostatistical interpolation (mean �15N : 7.69 ± 0.90 ‰)”.

7. Line 186: I understood why you used 5 size class instead of quantitative body size data. Anyway, it may be helpful for readers to include some justifications about the use of size class in the text.

In the text, we include some justifications about the use of size class

L. 196-202 (on the manuscript without track changes): “Individual effects of these two explanatory variables, as well as their interactions were examined. Using discrete size classes in isotopic analyses GLMs was justified a posteriori by the non-monotonic character of the isotopic ratios versus size relationships. For stomach contents analyses, size-class analyses allow to account for the highly non-linear nature of dietary mechanisms. Each individual within a size class is considered as a replicate of the class. Having the diet of several individuals within a size-class is required to capture the diet of a class, and to limit the importance of extreme feeding mechanisms”. 

8. Line 266, “… from -17.34 ± 0.68‰ (35+) to -16.77 ± 0.68‰ and -16.77 ± 0.65‰ (20-30cm) in winter.”: This sentence is confusing.

The sentence was modified for a better understanding: 

L. 282-283 (on the manuscript without track changes): « from -17.34 ± 0.68‰ (35+) to -16.77 for both 20-25 (-16.77 ± 0.68‰) and 25-30 (-16.77 ± 0.65‰) size classes in winter.”

9. Lines 294-296: Again, did you extract lipids from whiting samples? If not, the interpretation of their 13C value is difficult because the difference of lipid concentration may explain the ontogenetic and seasonal difference of 13C values.

We have not extracted the lipids for the reason stated above (see answer number 5). 

Moreover, as C:N ratios are similar for both seasons (3.18 ± 0.07 in autumn and 3.21 ± 0.09 in winter) and among size classes, influence of lipids on seasonal or ontogenetic differences can be considered negligible. Actual C:N ratios were missing and were added in the Table 1. 

10. Fig. 4. Legends of 13C and 15N are reversed.

Corrected 

Reviewer #2: The authors correctly addressed previous reviewers’ comments, and produced a sound manuscript that is acceptable for publication in PLOS one.

Below, I note a few errors:

1. Figure 3 is mislabeled: The left panel is 13C, whereas the central panel is 15N

Done

2. L. 198: change “analyzes” into analyses

Done

3. L267: change “increases” into increased

Done

7. PLOS authors have the option to publish the peer review history of their article (what does this mean?). If published, this will include your full peer review and any attached files.

Do you want your identity to be public for this peer review? For information about this choice, including consent withdrawal, please see our Privacy Policy.

Reviewer #1: No

Reviewer #2: No

---

## [Editor Report · Decision Letter 2]

7 Sep 2020

Seasonal and ontogenetic variation of whiting diet in the Eastern English Channel and the Southern North Sea

PONE-D-20-06925R2

Dear Dr. Timmerman,

We’re pleased to inform you that your manuscript has been judged scientifically suitable for publication and will be formally accepted for publication once it meets all outstanding technical requirements.

Kind regards,

Takefumi Nakazawa

Academic Editor

PLOS ONE
---

## [Editor Report · Acceptance letter]

11 Sep 2020

PONE-D-20-06925R2 

Seasonal and ontogenetic variation of whiting diet in the Eastern English Channel and the Southern North Sea 

Dear Dr. Timmerman:

I'm pleased to inform you that your manuscript has been deemed suitable for publication in PLOS ONE. Congratulations! Your manuscript is now with our production department. 

Kind regards, 

on behalf of

Dr. Takefumi Nakazawa 

Academic Editor

PLOS ONE